# Shadow Detection and Compensation from Remote Sensing Images under Complex Urban Conditions

**Tingting Zhou** [1] , **Haoyang Fu** [2,3], **Chenglin Sun** [1,4,*] **and Shenghan Wang** [1,4]

1    Coherent Light and Atomic and Molecular Spectroscopy Laboratory, College of Physics, Jilin University, Changchun 130012, China; ttzhou18@mails.jlu.edu.cn (T.Z.); shenghan@jlu.edu.cn (S.W.)
2    Department of Atmospheric and Oceanic Sciences & Institute of Atmospheric Sciences, Fudan University, Shanghai 200433, China; fuhaoyang@fudan.edu.cn
3    Zhuhai Fudan Innovation Research Institute, Zhuhai 519000, China
4    Key Laboratory of Physics and Technology for Advanced Batteries (Ministry of Education), College of Physics, Jilin University, Changchun 130012, China
*    Correspondence: chenglin@jlu.edu.cn; Tel.: +86-150-4307-6988

**Abstract:** Due to the block of high-rise objects and the influence of the sun's altitude and azimuth, shadows are inevitably formed in remote sensing images particularly in urban areas, which causes missing information in the shadow region. In this paper, we propose a new method for shadow detection and compensation through objected-based strategy. For shadow detection, the shadow was highlighted by an improved shadow index (ISI) combined color space with an NIR band, then ISI was reconstructed by the objects acquired from the mean-shift algorithm to weaken noise interference and improve integrity. Finally, threshold segmentation was applied to obtain the shadow mask. For shadow compensation, the objects from segmentation were treated as a minimum processing unit. The adjacent objects are likely to have the same ambient light intensity, based on which we put forward a shadow compensation method which always compensates shadow objects with their adjacent non-shadow objects. Furthermore, we presented a dynamic penumbra compensation method (DPCM) to define the penumbra scope and accurately remove the penumbra. Finally, the proposed methods were compared with the stated-of-art shadow indexes, shadow compensation method and penumbra compensation methods. The experiments show that the proposed method can accurately detect shadow from urban high-resolution remote sensing images with a complex background and can effectively compensate the information in the shadow region.

**Keywords:** shadow detection; shadow compensation; dynamic penumbra compensation method (DPCM); objected based

## 1. Introduction

Over the past 10 years, high-resolution remote sensing has been booming and prevailing in urban remote sensing [1,2]. However, medium and high-resolution remote sensing images are particularly affected by shadows, especially in urban areas with dense tall buildings, which leads to serious information loss for remote sensing images in the shadow region [3]. Accurate shadow extraction and information restoring in shadow areas can not only be used for the three-dimensional reconstruction of buildings, but also contribute great significance to urban feature classification, urban planning, road network extraction, and impervious layer research [4–6], etc. While working on shadow detection and information compensation, researchers mainly focus on RGB images, single shadow, or video sequences [7,8]. The remote sensing images, however, are prone to aerosol and sensor noise, which further complicates related shadow research. This is the reason why commonly used algorithms are absent from shadow detection and compensation for remote sensing images with a complex background and multiple shadows [9–11].

We refer to the entire process of shadow detection and compensation as shadow removal. Accurate shadow extraction is an important prerequisite for shadow compensation. Currently, scholars have proposed plentiful shadow extraction methods, including model-based, feature-based and machine learning-based ones [12]. The model-based methods acquire the shadow information by mathematical models constructed by the sensor's position and solar azimuth [3], in which the digital earth model (DEM) is the most typical one. For example, the DEM was employed in [13], where the author first obtained a rough shadow by DEM and applied SVM (Support Vector Machine) to optimize shadow feature. Nevertheless, the largest drawback of the model-based model is that it requires prior knowledge like camera and illumination orientation information, which are less accessible [14]. The feature-based methods usually involve image feature extraction and segmentation, where the feature could be combined with spectral, texture and semantic information [7,15], such as the shadow indexes based on color spaces for shadow highlighting. This type of algorithm mainly converts images from RGB model to hue and saturation intensity (HIS) space or the equivalent space like hue, saturation and value (HSV), hue, chroma and value (HCV), luma, inphase and quadrature (YCbCr and YIQ), in which shadow areas show higher hue and lower intensity than non-shadow areas [16]. Based on this, Tsai et al. [17] compared and analyzed the attribute of $(H_e + 1)/(I_e + 1)$ in different color spaces and obtained a shadow mask with Otsu's algorithm. Silva et al. [18] converted images to the CIELCh model, proposed a modified shadow index, and applied multilevel thresholding to obtain the shadow mask. Later, Ma et al. [19] improved Tsai's index and proposed NSVDI in the HSV space. However, the HSV space is restricted by the invalid values in the event of equal pixel values in the $R$, $G$ and $B$ bands—a ubiquitous situation among images. Compared with ordinary ones, multi-spectral remote sensing images often include NIR bands with longer wavelength, which are highly sensitive to shadows. Consequently, NIR is usually used in conjunction with other bands to strengthen shadow features [20], such as those shadow feature enhancement methods combined with NIR bands designed by [15,21,22]. Since 2015, the deep learning networks have been developing rapidly, providing new inspiration for shadow extraction. A multitude of effective deep learning methods are devised for shadow detection and have achieved better results than traditional methods [23–27], but the current machine learning-related algorithms are primarily for ordinary images. As a result of illumination changes and satellite revisit cycles, it is difficult to obtain shadowed and unshaded images of the same location to generate shadow datasets for shadow detection from remote sensing images. To sum up, the methods proposed to date are only suitable for shadow extraction under specific conditions [28], thus further research is necessary on efficient shadow extraction methods for remote sensing images.

Accurate shadow extraction is the premise of shadow compensation, also the core and challenge in shadow removal. The shadow area is the part with information blocked in the images, and how to use the information in the non-shadow area to recover the shadow area is the key to shadow compensation. Common methods for shadow compensation are histogram-matching methods, linear correlation correction (LCC) methods, light intensity ratio-based methods, and machine learning-based methods, etc. Histogram-matching methods [29–32]: the histogram information of the non-shadow region is applied to match the shadow region and information compensation. LCC methods [4,33,34]: the reconstruction relies on a linear correlation function to compensate shadow regions by adjusting the intensities of the shadow pixels according to the statistical characteristics of the corresponding non-shadow regions. The intensity ratio-based methods [18,35]: these methods consider that the shadow region is formed by the ambient light and that the non-shadow region consists of ambient light and direct light, and lit the shadow by the ratio of direct light to ambient light. The machine-learning based methods: previous machine learning methods were often dedicated to shadow detection, with the arduous recovery of shadow information. In recent years, various deep-learning frameworks, such as generative adversarial networks (GANs) [24,26,28], have been proven to be one of the most appropriate

frameworks for shadow removal. The GAN-based methods acquire the shadow-free image by generating G and discriminator D, where G is trained to produce a realistic image and D to distinguish the shadow-free images produced by G. In addition to considering which algorithm to use for shadow compensation, from where to select information is also a vital issue in shadow removal. Finlayson et al. [36,37] removed shadows by the gradient domain information from the boundary between shadow and non-shadow regions, and Zhang et al. [12] proposed an inner–outer outline profile line (IOOPL) algorithm according to the similarity between inner and outer pixels of boundary. The brightness of a landcover with or without shadow occlusion greatly changes, but its gradient and texture are generally constant. Based on these features [33,35], an object-based method with an object as the smallest processing unit was applied to find the most similar non-shadow object for each shadow object. However, in terms of remote sensing images with complex backgrounds and multiple shadows, efficient and accurate shadow compensation methods remain absent.

In addition to the shadow mask from shadow detection, there is also a penumbra region whose shadow characteristics gradually decrease from shadow to non-shadow. If penumbra was compensated in the same way as umbra, there will be "oversaturated" in penumbra. Many studies classify penumbra compensation as a "boundary" problem. For instance, Bauer et al. [38] proposed the edge smoothing algorithm using an average value from the boundary-surrounding area. Zhang et al. [39] employed a constrained texture synthesis to compensate the texture and illumination information on the penumbra region. Khan et al. [33] processed shadow objects on the boundary by boundary objects in the shadow contour with a minimum relative change in intensity. Nevertheless, the effects of these methods were insufficient, and how to accurately define penumbra and restore the dynamic attenuation value in penumbra remains a challenge for penumbra removal.

Based on the analysis above, we noticed that despite numerous related studies for shadow removal, shortcomings in shadow detection, shadow compensation and penumbra removal exist as follows: (1) even with a number of shadow detection and removal methods, most of the them are focused on single shadow removal but not applicable to multiple shadows; (2) many strategies are formulated for using non-shadow area information to compensate for shadow areas, but the selection of information recovery samples and algorithms to be used is difficult in related research; (3) processing penumbra by the strategies used in the edge smoothing/matting would be imprecise, as the penumbra value changes dynamically. Furthermore, there is a lack of algorithm to accurately define the range and effective removal of penumbra.

The main contributions of this paper are: (1) an improved shadow index (ISI) is proposed in combination with the characteristics in the YCbCr model and NIR bands in multi-spectral images; (2) according to the similarity of ambient illumination conditions among adjacent objects, a method is presented to restore the information of shadow objects through the nearest non-shadow objects; and (3) a dynamic penumbra compensation method (DPCM) is proposed to define the penumbra scope and compensate for penumbra.

The flowchart in Figure 1 shows the principal steps of the proposed methodology. In the shadow detection step, we first enhanced shadow features by an improved shadow index (ISI), which was design by the YCbCr model and NIR band. Meanwhile, the image was also segmented by a mean-shift algorithm and the objects from segmentation were applied to reconstruct the result from ISI. Then, the threshold segmentation was executed to obtain shadow masks. In the shadow compensation step, we presented an object-based shadow compensation method, based on the information from the neighbor non-shadow objects. After the objects from segmentation were treated as a minimum processing unit, we put forward a shadow compensation method which always compensates shadow objects with their adjacent non-shadow objects since the adjacent objects are most likely to have the same ambient light intensity. We also presented a dynamic penumbra compensation method (DPCM) to define the penumbra scope and remove the penumbra accurately.

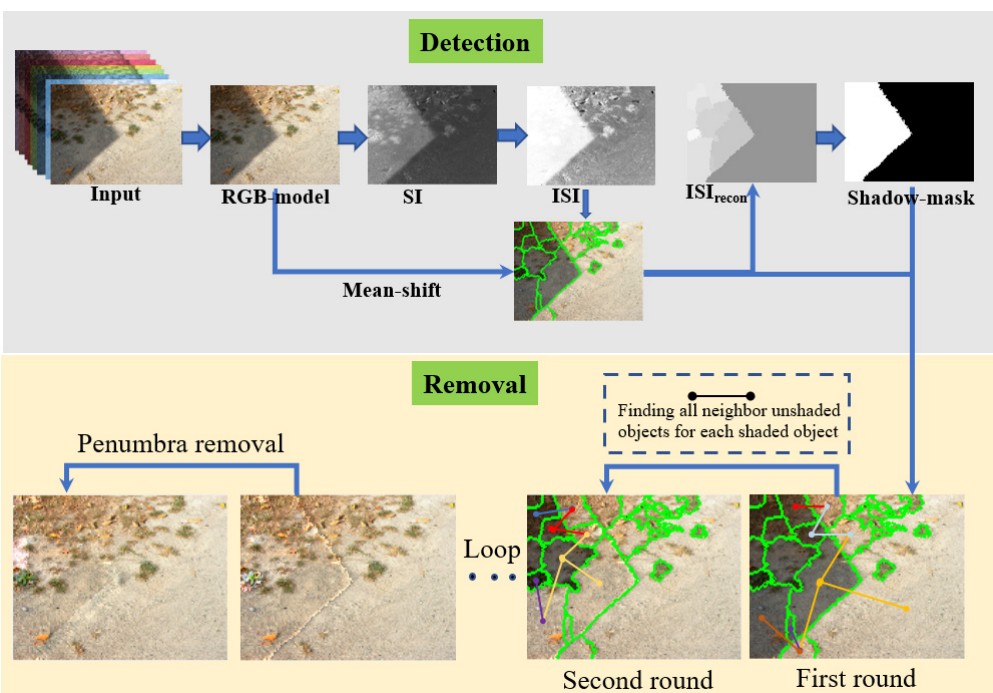

**Figure 1.** Flowchart of the proposed methods.

## 2. Shadow Detection

Shadows often appear in urban areas with tall buildings. Accurate shadow detection is essential for shadow removal, but also an important prerequisite for shadow compensation. According to different forms, shadows can be divided into the following types: the cast shadow is caused by objects blocking the sunlight; the self-shadow is a shadow that is formed in some parts of an object which are not directly illuminated by sunlight; the umbra arises from the totally blocked direct illumination, and the penumbra is formed when the direct illumination is partly blocked. The representation of shadows is shown in Figure 2.

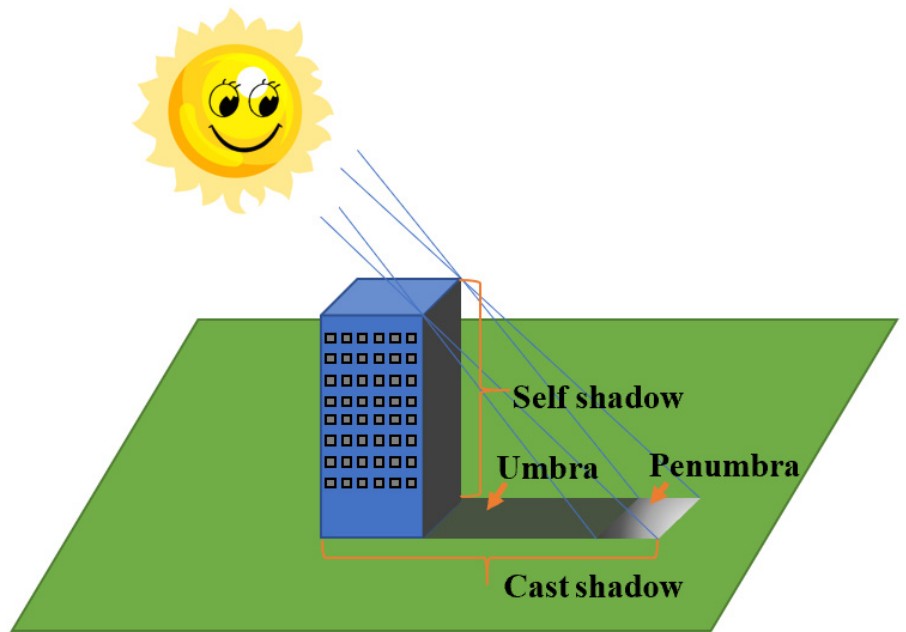

**Figure 2.** Representation of shadows.

## 2.1. Image Segmentation with Mean-Shift Method

An image from WorldView-3 was selected as the experimental data, which was taken on 8 March 2016, at 12:12 in Tripoli, Libya. The data were provided by Digital Global on the website (http://www.digitalglobe.com/samples, accessed on 9 December 2020) [40] as product samples. We processed the data with radiation calibration, Flaash atmospheric correction and image fusion to acquire an eight-band image with a resolution of 0.31 m. The details of the experimental image are listed in Table 1. For a more intuitive comparison between the proposed penumbra removal method and state-of-art methods, this paper selected the UIUC [35] dataset and the SRD [23] dataset containing both shadow and shadow-free images to conduct the qualitative analysis of the penumbra removal results. All the data in this paper were normalized to facilitate statistics and comparative analysis.

**Table 1.** Details of WorldView-3 imagery.

|  | **CoastalBlue** | **Blue** | **Green** | **Yellow** | **Red** | **RedEdge** | **NIR1** | **NIR2** |
|---|---|---|---|---|---|---|---|---|
| Central wavelength (nm) | 425 | 480 | 545 | 605 | 660 | 725 | 832.5 | 950 |
| Resolution | Multi-spectral resolution: 1.24 m Panchromatic resolution: 0.31 m Short-wavelength infrared: 3.7 m | | | | | | | |
| Location | Tripoli, Libya | | | | | | | |
| Local time | 8 March 2016, 12:12 | | | | | | | |

Mean-shift [41] is a feature space analysis algorithm widely used in image segmentation. Unlike those methods requiring prior knowledge of the clusters number, the mean-shift algorithm is a nonparametric clustering technique. With the application of mean-shift, homogeneous pixels with similar spectral and texture features can be unified. In shadow detection, the objects from segmentation were used to reduce the interference of high reflectance landcovers in the shadow region and optimize the edge features of shadow masks. In shadow compensation, the objects were also used as the minimum processing unit in shadow compensation. After the comparative analysis, we set the segmentation spatial radius as 9, and the segmentation feature space radius as 15, and the minimum segment area as 200 in the mean-shift.

When using the mean-shift algorithm, the spatial resolution of the image is proportional to the parameter size setting of the mean-shift. High reflectance landcovers such as white vehicles, etc., interfere with shadow extraction. If these high reflectance landcovers can be averaged with other surrounding shadow pixels, the impact of high reflectance landcovers on shadow extraction will be greatly reduced. In the remote sensing image of urban areas, high reflectance vehicles and small noises are the main cause of obvious interference to shadow detection. Taking WorldView-3 data as an example, the size of the vehicle is about 4.5 m × 1.5 m (the spatial resolution of the image is 0.31 m, one vehicle occupies about 75 pixels), and we set the minimum segment area above 150, so the high reflectance vehicle will be merged with the surrounding shadow pixels into one object, and the impact of high reflectance vehicle will be reduced. After comparative analysis, we set the segmentation spatial radius as 9, the segmentation feature space radius as 15 and the minimum segment area as 200 in the mean-shift.

## 2.2. Improved Shadow Index (ISI)

Various shadow indexes based on different color spaces have been proven to be able to effectively enhance shadow features, such as [17] Tsai who compared the effect of shadow enhancement of $(H_e + 1)/(I_e + 1)$ in different color spaces, including the HSI, HSV, HCV, YIQ, and YCbCr models. According to the results analysis, we found that Tsai's method in the YCbCr model ($T_{YCbCr}$) can enhance shadows better than other models. YCbCr is a family of color spaces used as a part of the color image pipeline in video and

digital photography systems, where *Y* is the luma component; and *Cb* and *Cr* are the blue-difference and red-difference chroma components, respectively.

$$
\begin{bmatrix} Y \\ Cb \\ Cr \end{bmatrix} = \begin{bmatrix} 0.257 & 0.504 & 0.098 \\ -0.148 & -0.291 & 0.439 \\ 0.439 & -0.368 & -0.071 \end{bmatrix} \begin{bmatrix} R \\ G \\ B \end{bmatrix} + \begin{bmatrix} 16 \\ 128 \\ 128 \end{bmatrix} \tag{1}
$$

In Equation (1), the *Y*, *Cb*, and *Cr* components in the YCbCr model are identical or equivalent to the intensities of component *I*, saturation component *S* and hue component *H* in the HSI model [17], respectively. Compared with the non-shadow areas, the shadow areas have lower intensity, as the electromagnetic radiation from the sun is blocked. Meanwhile, the saturation with a short blue-violet wavelength in shadow areas is higher, which is caused by the Rayleigh effect of atmospheric scattering [42]. Under these conditions, we know the higher *Cb* and lower *Y* in the shadow pixels. The change from Figure 3a to Figure 3b in the YCbCr space is that the shadow area turns into green, which shows that the *Cb* information is obviously enhanced and *Y* is weakened significantly in the shadow area. In light of the above analysis, we defined SI as

$$
\mathrm{SI} = \frac{C_b - Y}{C_b + Y} \tag{2}
$$

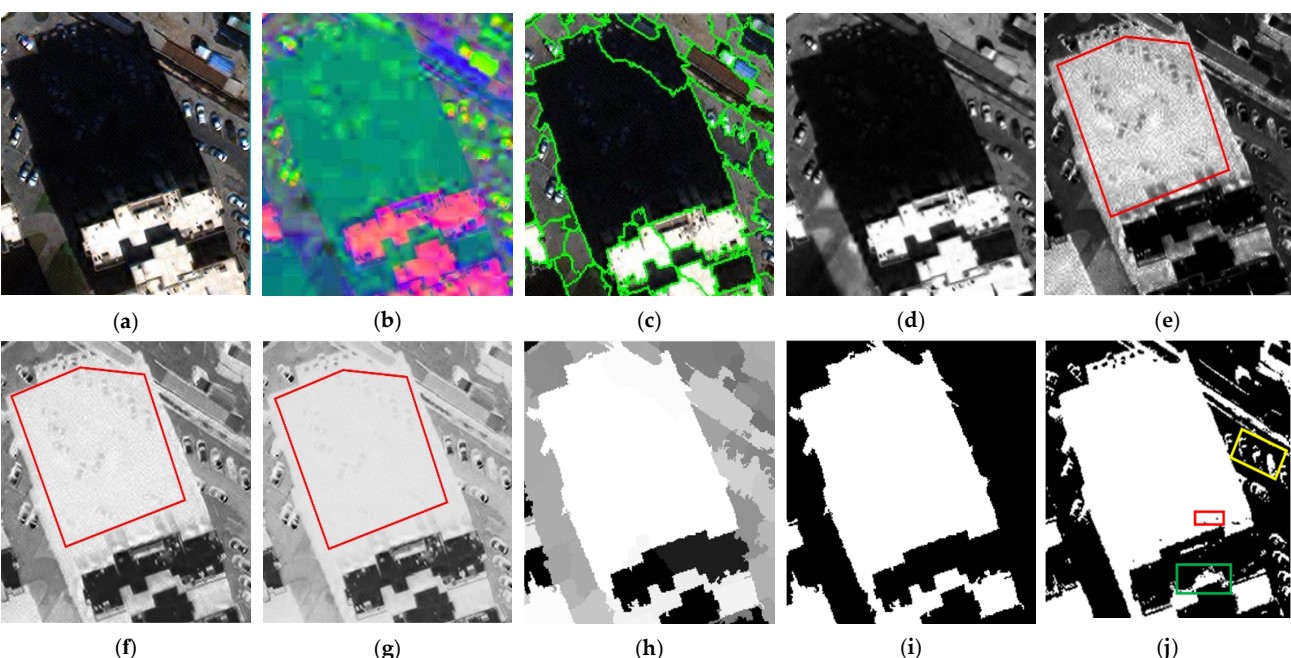

**Figure 3.** Shadow extraction process, where (**a**) is the original image displayed by RGB bands; (**b**) is YCbCr model; (**c**) is the boundary of the mean-shift segmentation result; (**d**) represents the NIR band; (**e**) is the $T_{YCbCr}$; (**f**) is the result of SI; (**g**) is the result of ISI; (**h**) is the reconstructed ISI by using segmentation results; (**i**) is the threshold segmentation result of (**h**); and (**j**) is the result of threshold segmentation of (**g**).

However, a color model like YCbCr is greatly sensitive to color, based on which SI proposed is prone to the high reflectance landcovers such as white vehicles in the shadow region (see Figure 3f).

Compared with ordinary images, multi-spectral images provide richer spectral information, especially in near-red bands [22,43], which have been proven as highly sensitive to shadows and are common in shadow-related research. After the spectral statistical analysis of various landcovers, we found that the energy of the shadow region in NIR is notably

lower than that of the non-shadow region. Therefore, for the multi-spectral images with the NIR band, we introduced NIR to improve SI and the proposed ISI:

$$ISI = \frac{SI + (1 - NIR)}{SI + (1 + NIR)} \tag{3}$$

The result of ISI is shown in Figure 3g, from which we can intuitively see that compared with SI, the characteristics of high reflectance landcovers in the shadow region are obviously weakened in ISI. Figure 3e is the result of $T_{YCbCr}$, where the vehicles in the shadow with high reflectance influence the result, though the shadows have been highlighted.

As shown by the red polygons in Figure 3e–g, we expect that the value of the shadow area tends to be consistent and can be distinguished from other landcovers. However, it is clear in Figure 3e–f that the value of vehicles in the shadow area is relatively low. If segmentation is directly executed in SI or $T_{YCbCr}$, the location of these vehicles can easily be judged as non-shadow.

We performed histogram statistics on the results of $T_{YCbCr}$, SI and ISI and the shadow area in the red polygons in Figure 3e–g, and the statistical results are shown in Figure 4. For $T_{YCbCr}$, Figure 4d plots that the value in the red polygon in Figure 3e is greater than 0.3, whereas Figure 4a manifests the difficulty in finding a clear threshold to distinguish the high reflectance landcovers in a shadow area from the low reflectance landcovers in a non-shadow area. From Figure 4b to Figure 4c, the value of the shadow is concentrated from 0.4–0.8 to 0.7–0.8, and the high-reflectance landcovers in the shadow area have been well enhanced (see Figure 3f,g).

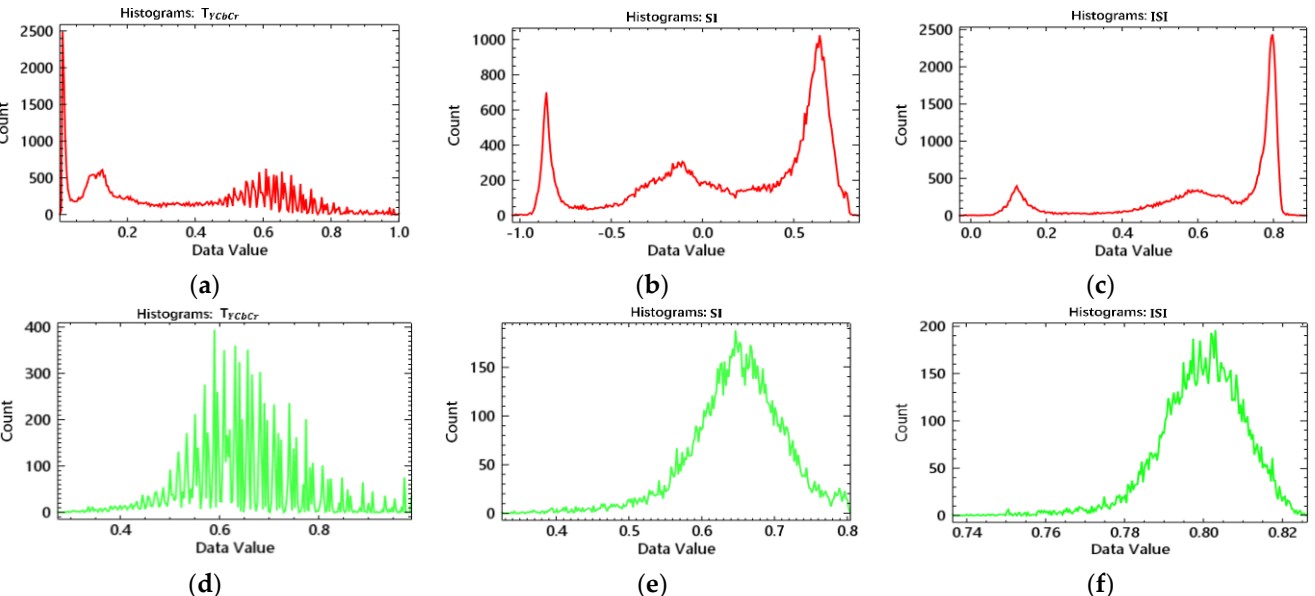

**Figure 4.** Histogram statistic: (**a**–**c**) are the results of Figure 3e–g and (**d**–**f**) are the results of the red polygons in Figure 3e–g. The first column is the result from $T_{YCbCr}$; the second column is the result of SI; and the third column is the result of ISI.

### 2.3. Reconstruction of ISI and Shadow Classification

The purpose of accurate shadow detection is to recover information in shadow areas, so that we can count and analyze the objects blocked by shadow. In actual application, the main cause of information loss in remote sensing images is the block of huge man-made features such as buildings and viaducts, while the shadows of pedestrians and vehicles can be ignored. For the obtained ISI, if segmentation is directly applied to ISI, small noise will be serious, and the integrity of the shadow edge will be adversely affected. As shown in Figure 3j, the green rectangle is the error caused by the high reflectance landcover in the shadow area; the red one is the shadow edge lacking integrity; and the yellow rectangle is

unnecessary shadow information. To reduce these errors, we used the segmentation results from the mean-shift to reconstruct ISI by replacing the values of pixels in each object with its average value. Assuming that the total number of segmented objects is $n$, and each corresponding object contains $k_i$ pixels, $1 \leq i \leq n$, then the reconstituted $\text{ISI}_{\text{recon}}$ is:

$$\text{ISI}_{\text{recon}} = \bigcup_{i=1}^{n} \left( \left( \sum_{j=1}^{k_i} V_j \right) / k_i \right) \tag{4}$$

where $V$ is the value of pixel $j$. The results of ISI and reconstruction are listed in Figure 3g,h, from which we can see that the small shadows formed by vehicles and the surrounding non-shadow pixels are integrated into one object, and the shadow characteristics are improved.

For the reconstructed ISI, the threshold from the histogram statistic was applied to obtain the shadow mask. The shadow mask is shown in Figure 3i, where the edge characteristics of the shadow are well preserved with small shadows properly removed.

## 3. Shadow Compensation and Post-Processing

For the multi-spectral image, this paper restored the shadow information in RGB bands. For any object, the adjacent objects are likely to have similar ambient illumination. We used a method based on the combination of the illumination intensity ratio and the nearest adjacent non-shadow objects to compensate the shadow mask. We assume that the illumination is composed of ambient illumination and direct illumination. In a real shadow scene, the cast shadow is composed of penumbra and umbra. As the umbra is only composed of ambient illumination, the penumbra consists of ambient illumination and gradually changing direct illumination. Based on this, the shadow compensation can be divided into the shadow compensation and the post-processing of penumbra removal including penumbra detection and compensation.

### 3.1. Shadow Compensation Based on the Information from Adjacent Objects

In the image formation theory from [44], for any pixel $i$ in the image, the value $I_i$ of $i$ can be expressed as the product of the pixel's illumination intensity $L_i$ and the reflectivity $R_i$:

$$I_i = L_i \cdot R_i \tag{5}$$

The illumination consists of direct illumination and ambient illumination [39]: the direct illumination originates directly from solar radiation, and the ambient illumination is mainly caused by sky scattering. Therefore, we assumed that the shadow region is formed by ambient illumination accompanied by partial or no direct illumination, whereas the non-shadow region involves both direct and ambient illuminations. From Equation (5), the value of any pixel can be expressed as

$$\begin{aligned} I_i^{\text{non-shadow}} &= (L_i^a + L_i^d) R_i \\ I_i^{\text{shadow}} &= (L_i^a + \alpha L_i^d) R_i \ \alpha \in [0, 1) \end{aligned} \tag{6}$$

where $L_i^a$ and $L_i^d$ is the ambient and direct illumination of the pixel $i$, respectively. $\alpha$ is the attenuation factor of the direct illumination, and $\alpha = 1$ means the pixel in the non-shadow region; $\alpha = 0$ means the pixel in umbra; $\alpha \in (0, 1)$ means the pixel in penumbra.

Equation (6) reveals that, for a pixel $i$, the ratio $r$ of the ambient illumination intensity and the direct illumination intensity can be denoted as

$$r = \frac{I_i^{\text{non-shadow}} - I_i^{\text{shadow}}}{I_i^{\text{shadow}}} = \frac{(L_i^a + L_i^d) - (L_i^a + \alpha L_i^d)}{(L_i^a + \alpha L_i^d)} \tag{7}$$

Based on Equation (7), the value of a pixel $i$ with shadow removal can be expressed as

$$I_i^{\text{non}-shadow} = \frac{(r+1)}{(\alpha r + 1)} I_i \tag{8}$$

where for pixel $i$, $\alpha = 1$ means in a non-shadow region; $\alpha = 0$ represents in umbra, or in penumbra when $\alpha \in (0,1)$. For the processing in this section, $\alpha = 0$, and all the shadow masks obtained are processed as umbra.

In shadow compensation, the objects segmented in Section 2 were used as the minimum processing unit. If the direct light is not blocked, each object and its neighboring objects receive the most similar solar radiation. Therefore, using the information of non-shadow objects adjacent to a shadow object is the most accurate strategy to compensate the shadow object. Thus, the schematic diagram of shadow compensation is exhibited in Figure 5 and the shadow compensation steps are as follows:

1. Mark the shadow region as $S$ containing $m$ objects, and label all the non-shadow regions as $U$ with $n$ objects;
2. Find all the shadow objects directly adjacent to the non-shadow region labeled as $S^{\text{near}}$ with $f$ objects;
3. Describe a shadow object in $S^{\text{near}}$ as $S_i^{\text{near}}$, $i \in \{1, 2, 3, \cdots f\}$, find all non-shadow objects adjacent to $S_i^{\text{near}}$, and mark them as $U_i^{\text{near}}$ containing $k$ objects;
4. For the $i$th shadow object $S_i^{\text{near}}$, its average value in band $q$ is $S_{i,q}^{\text{avg}}$, $q \in \{R, G, B\}$, and the average value of $k$ adjacent non-shadow objects $U_{i,j,q}^{\text{near}}$ in band $q$ is $U_{i,j,q}^{\text{avg}}$, $j \in \{1, 2, 3, \cdots k\}$. $S_{i,q}^{\text{avg}}$ and $U_{i,j,q}^{\text{near}}$ are used to calculate the ratio of the direct light intensity to the ambient light intensity, then to get the mean value of the $k$ ratios:

$$r_q = \left(\sum_{j=1}^{k} \left(\frac{U_{i,j,q}^{\text{avg}} - S_{i,q}^{\text{avg}}}{S_{i,q}^{\text{avg}}}\right)\right) \cdot \frac{1}{k} \tag{9}$$

5. Then $S_{i,q}^{\text{avg}}$($\times M$ pixels) treated with shadow compensation in band $q$ is $U_{i,q}^{\text{near,removal}}$:

$$U_{i,q}^{\text{near,removal}} = \bigcup_{l=1}^{M} \left((r_q + 1)S_{l,i,q}^{\text{near}}\right) \tag{10}$$

The $S_i^{\text{near}}$ after shadow compensation is $U_i^{\text{near,removal}}$:

$$U_i^{\text{near,removal}} = \bigcup_{q=R,G,B} U_{\text{removal},i,q}^{\text{near}} \tag{11}$$

6. Repeat Step 3 to 5 until $i = f$, and the $S^{\text{near}}$ after shadow compensation is:

$$U^{\text{near,removal}} = \bigcup_{i=1}^{f} U_i^{\text{near,removal}} \tag{12}$$

7. Update $S$, $U$, $m$ and $n$: $S = S - S^{\text{near}}$, $U = U + U^{\text{near,removal}}$, $m = m - f$, and $n = n + f$;
8. Loop Steps 2 to 7, until $m = 0$, then the shadow compensation process is complete.

### 3.2. Dynamic Penumbra Compensation Method (DPCM)

After processing the shadow compensation for the shadow mask, we found that the pixels in the junction of the shadow and non-shadow were "oversaturated", as shown in Figure 6b. This was because the objects after image segmentation were used as the smallest processing unit, and the same shadow compensation strategy was applied to compensate information in each object. However, among the shadow objects directly adjacent to the non-shadow objects, there is a penumbra zone in the transition zone between shadow and

non-shadow. If the method ($\alpha = 1$) in Section 3.1 was utilized for penumbra removal, "oversaturation" will occur.

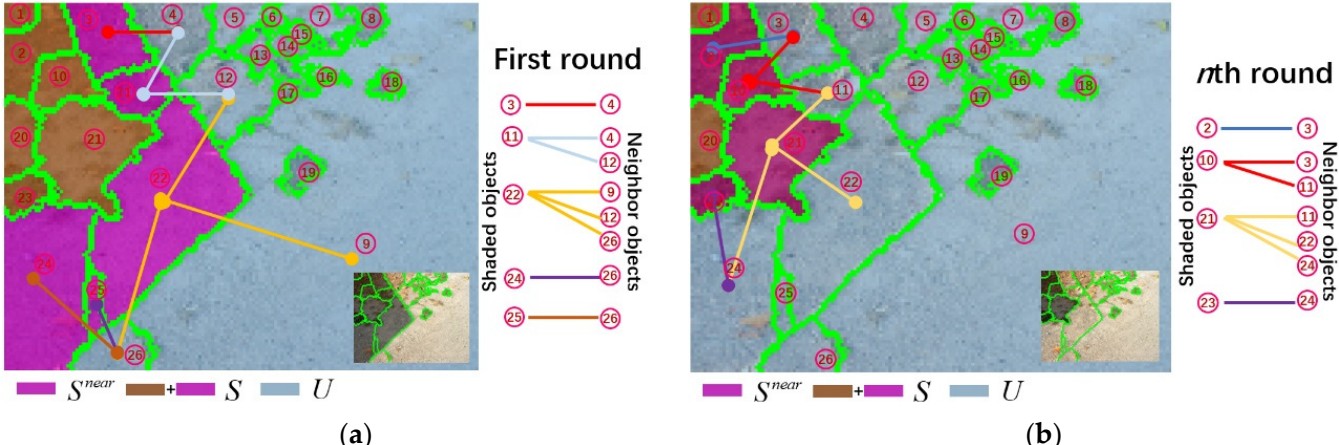

**Figure 5.** Schematic diagram of shadow compensation using adjacent objects information: (**a**) on the first round, the non-shadow objects directly adjacent to shadow objects were used to compensate their neighbor shadow objects; (**b**) on the *n*th round (here *n* = 2), the non-shadow objects executed shadow compensation on the ($n-1$)th and were utilized for shadow objects compensation.

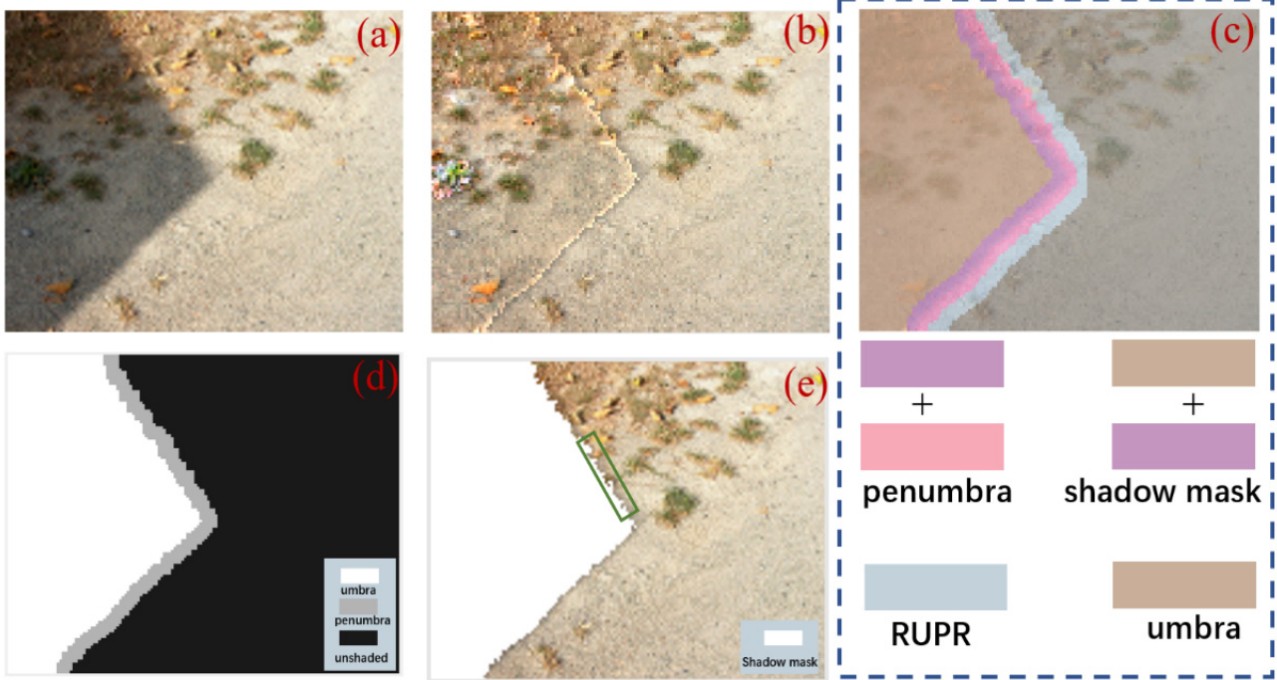

**Figure 6.** Penumbra removal: (**a**) original image; and (**b**) the result of shadow compensation; (**c**) the region sampled for penumbra removal; (**d**) shadow sketch map: white for umbra, gray for penumbra and black for unshaded region; and (**e**) shadow mask obtained in shadow detection.

During the shadow extraction process, we cannot guarantee whether the obtained shadow mask includes all penumbra pixels. As shown in the green box in Figure 6e, the shadow mask did not include all the penumbra pixels. The experimental statistics reveal that the penumbra is generally concentrated within 10 pixels width, and the shadow mask can only include the majority of the penumbra pixels but not all. For the small part, penumbra pixels with extremely weak shadow characteristics, the shadow extraction algorithm cannot identify them as a shadow. Therefore, in this article, we use the new

shadow obtained by eroding the shadow mask by seven pixels width as the umbra (see Figures 6c and 7). Then, the new obtained umbra was dilated by 10 pixels width to gain the penumbra region (see Figures 6c and 7). After that, we dilated the penumbra outwards with five pixels width, which was treated as the region used for penumbra removal (RUPR), as illustrated in Figure 6c,d.

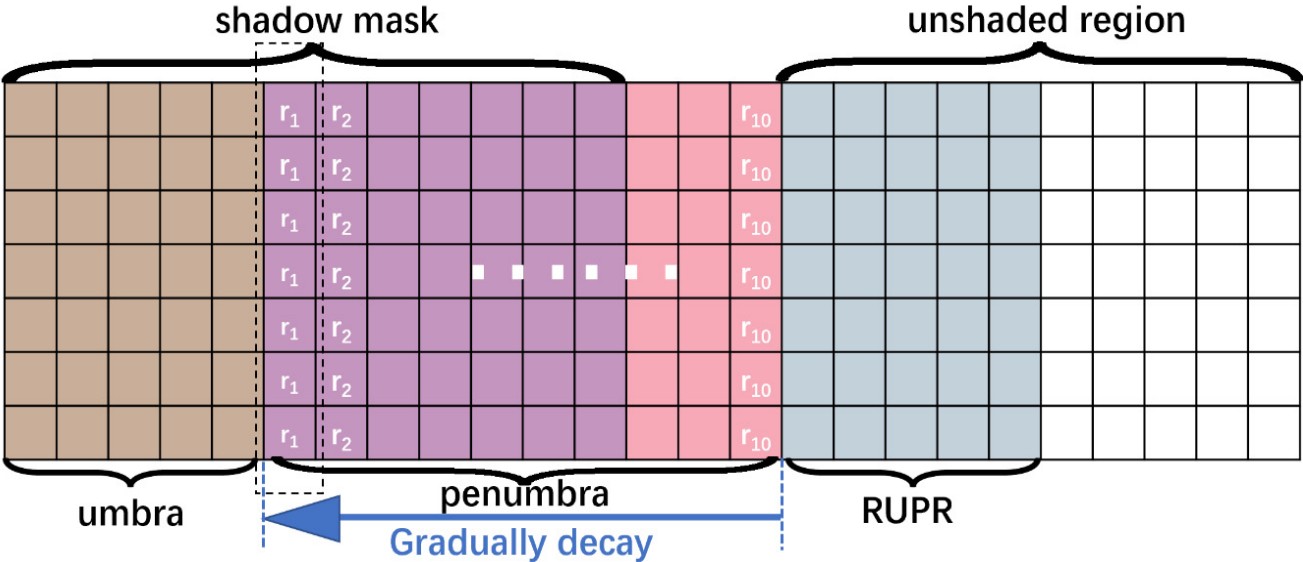

**Figure 7.** Schematic calculation of $r$. For each one pixel width region in penumbra, the ration $r$ was updated.

According to Equation (6), the attenuation coefficient $\alpha \in (0,1)$, any pixel $i$ in the penumbra can be expressed as

$$I_i^{\text{penumbra}} = (L_i^a + \alpha L_i^d)R_i \; \alpha \in (0,1) \tag{13}$$

The direct light intensity gradually decreases from the non-shadow region to the umbra direction, i.e., $\alpha$ gradually decreases. As shown in Figure 7, we believe that along the decay direction, the degree of attenuation is the same within each pixel width area. For the $n$th, ($n = 1, 2, 3, \cdots 10$) single pixel width area in penumbra, the ratio of penumbra and non-shadow region in band $q$, $r_{n,q}$ is:

$$r_{n,q} = \frac{I_q^{\text{RUPR}} - I_{n,q}^{\text{pen}}}{I_{n,q}^{\text{pen}}}, \; q \in \{R, G, B\} \tag{14}$$

where $I_{\text{RUPR},q}$ represents the average value of the RUPR in band $q$; $I_{n,\text{pen},q}$ denotes the average value of the $n$th single-pixel width area in penumbra in band $q$.

For pixel $i$ in the $n$th width of penumbra, its value with penumbra removal can be represented as

$$I_{n,i,q} = (r_n + 1)I_q^{\text{RUPR}}, \; q \in \{R, G, B\} \tag{15}$$

where $I_{n,i,q}$ stands for the pixel after penumbra removal.

Note that, in order to process all pixels in the penumbra, the 10 pixel width penumbra acquired by the proposed method is larger than or equal to the range of the actual penumbra. The process of DPCM is represented in Figure 8. For each step, a new ratio $r$ was calculated to recover the penumbra information in this pixel width region.

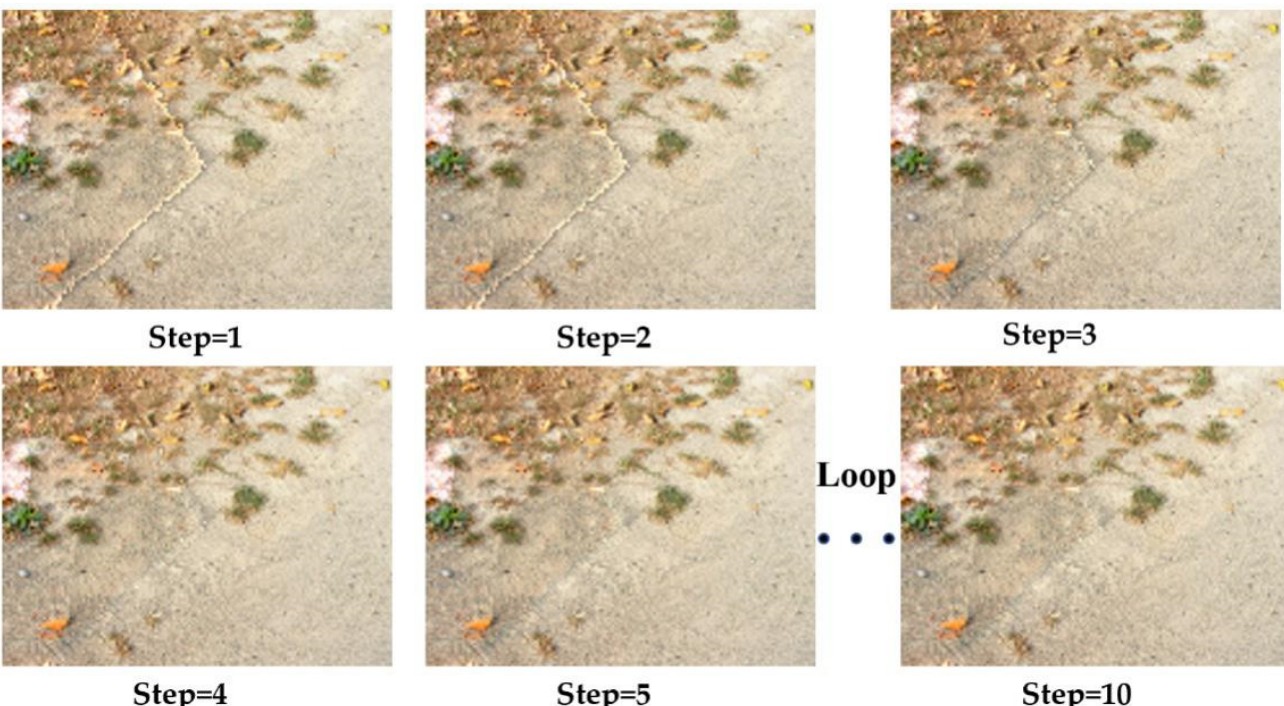

**Figure 8.** Process of dynamic penumbra compensation. From Steps 1 to 10, the ratio *r* was calculated separately and the penumbra was compensated step by step.

## 4. Results

In this section, several experimental sites were selected from the WorldView-3 data to test the proposed shadow extraction, shadow compensation and penumbra removal methods. In order to prove the performance and robustness of the proposed algorithm, we compared the experimental results with the stated-of-art shadow indexes, shadow compensation methods and penumbra removal methods.

### 4.1. Preference and Comparative Analysis of Shadow Detection

4.1.1. Qualitative Analysis of Shadow Detection

Two sites were selected for shadow detection: Site 1 contains many tall buildings and bare soil, with a large shadow covering area, which leads to a serious information loss in the shadow area. When it is used for shadow detection analysis, the integrity of the shadow edge and the degree of separation from other low reflectance landcovers can be well observed; Site 2 covers a lot of complex surfaces and has plentiful fragmentary shadows, which can well reflect the accuracy of shadow extraction. The results of SI and ISI (see Figure 9b,c and Figure 10b,c) demonstrate that, with the assistance of the NIR band, the darker regions caused by the high reflectance landcovers in the shadow area are well improved. Large-area shadows caused by the occlusion of buildings and viaducts need to be removed because of the serious loss of feature information in the shadow area, while the shadows formed by small ground features such as vehicles and shrubs can be ignored. With the application of the reconstruction method in Section 2.3, noises and small shadow region are removed, as exhibited in Figures 9d and 10d.

4.1.2. Comparative Analysis of Shadow Detection and Accuracy Evaluation

To verify the superiority of the proposed shadow extraction method, we compared ISI with the state-of-art shadow indexes and evaluated their shadow extraction accuracy. The indexes subject to the comparative analysis were the combinational shadow index (CSI) [15], the shadow detection index (SDI) [18], the normalized saturation-value difference index

(NSVDI) [19] and Tasi's method [17] in YCbCr (T$_{YCbCr}$) and HSV T$_{HSV}$ space, respectively. They are specified in Table 2.

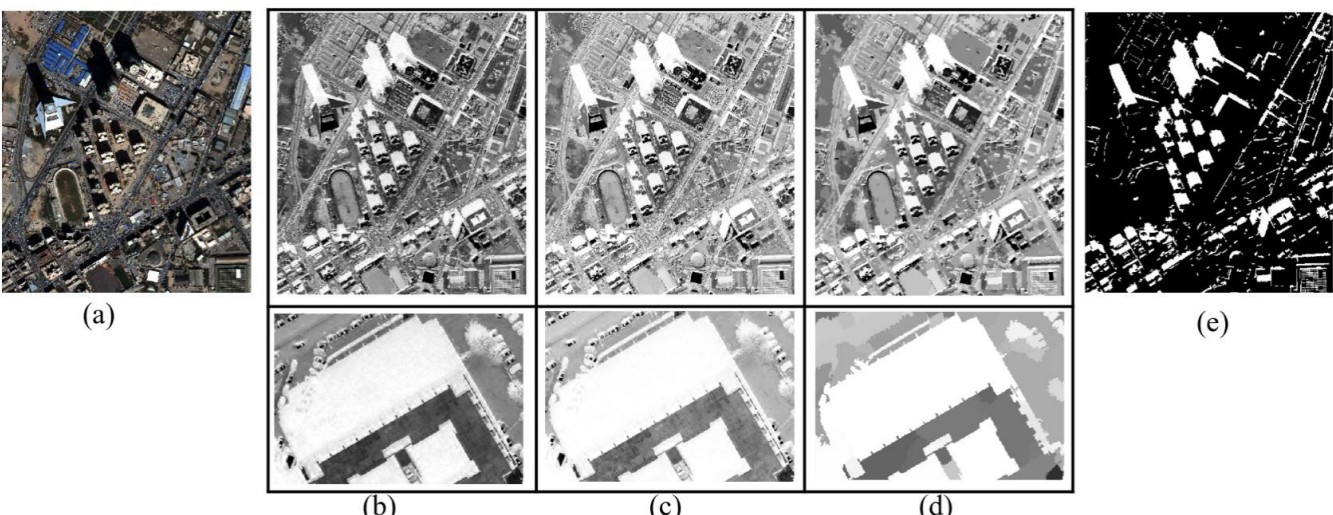

**Figure 9.** Experimental site 1: (**a**) original image shown with RGB bands; the first rows of (**b**–**d**) are the results of SI, ISI and the reconstructed ISI, respectively; the second rows of (**b**–**d**) are the local enlarged view of SI, ISI and the reconstructed ISI, respectively; and (**e**) the result of shadow detection after threshold segmentation.

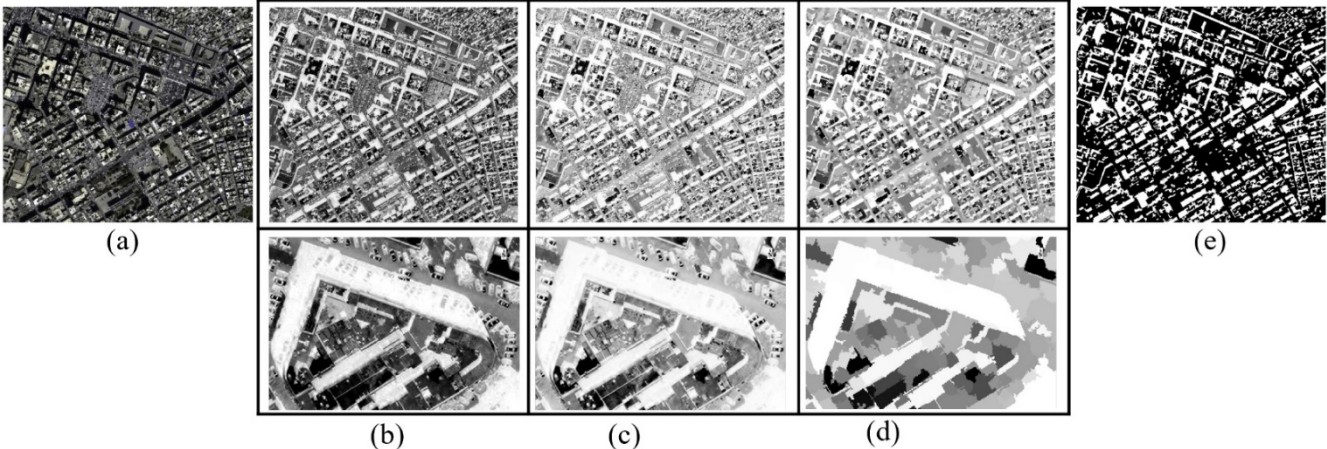

**Figure 10.** Experimental site 2: (**a**) original image shown with RGB bands; (**b**) result of shadow detection after threshold segmentation; the first rows of (**c**–**e**) are the results of SI, ISI and the reconstructed ISI, respectively; the second rows of (**c**–**e**) are the local enlarged view of SI, ISI and the reconstructed ISI, respectively.

**Table 2.** The definitions of the indexes.

| Index | Data | Definition |
|---|---|---|
| CSI | Sentinel-2A | $SEI = \frac{(b_{443nm}+b_{945nm})-(b_{560nm}+b_{842nm})}{(b_{443nm}+b_{945nm})+(b_{560nm}+b_{842nm})}$ <br> $CSI = \begin{cases} SEI - NIR, if\ NIR \geq NDWI \\ SEI - NDWI, else \end{cases}$ |
| SDI | WorldView-2 | $SDI = \frac{NIR2-Blue}{NIR2+Blue} - NIR1$ |
| NSVDI | IKONOS | $NSDVI = \frac{S-V}{S+V}$ executed in HSV space |
| T$_{YCbCr}$, T$_{HSV}$ | RGB data | Executed in HSV and YCbCr models separately |

The calculation results of each index are shown in Figures 11 and 12. To better observe the details of each index, the results in Figure 12 are partially enlarged in Figure 13. Figures 11 and 12 reveal that NSVDI, CSI, and $T_{YCbCr}$ can all enhance the shadow region, but suffer from their own shortcomings compared with the results of ISI. The HSV space is restricted by the fact that when the pixel values in *R*, *G* and *B* bands are equal, the denominator of the HSV's definition is 0, generating invalid values. Hence, the enlarged view of NSVDI in Figure 13 manifests that although the region marked by the red ellipses are in the shadow region, they were calculated as invalid values. There are many asphalt roads in Site 1 whose properties are similar to shadow, and CSI fails to distinguish shadow from these low reflectance areas; however, even CSI can obtain similar results to ISI. From Figures 11d and 12d, we can see that one can illuminate the shadows and distinguish them from the low reflectance landcovers. However, as illustrated by the local enlarged figure of Figure 13, these are greatly disturbed by the high reflectance landcovers in the shadow region. If threshold segmentation is carried out directly, it causes omission errors. In addition, Figures 11e and 12e depicts shadows that cannot be distinguished from low reflectance objects, resulting in a large commission error. From the result in Figure 11f, the SDI result cannot distinguish shadows from low-reflectance objects, nor can it distinguish them from vegetation, which leads to both high commission error and omission error in shadow extraction.

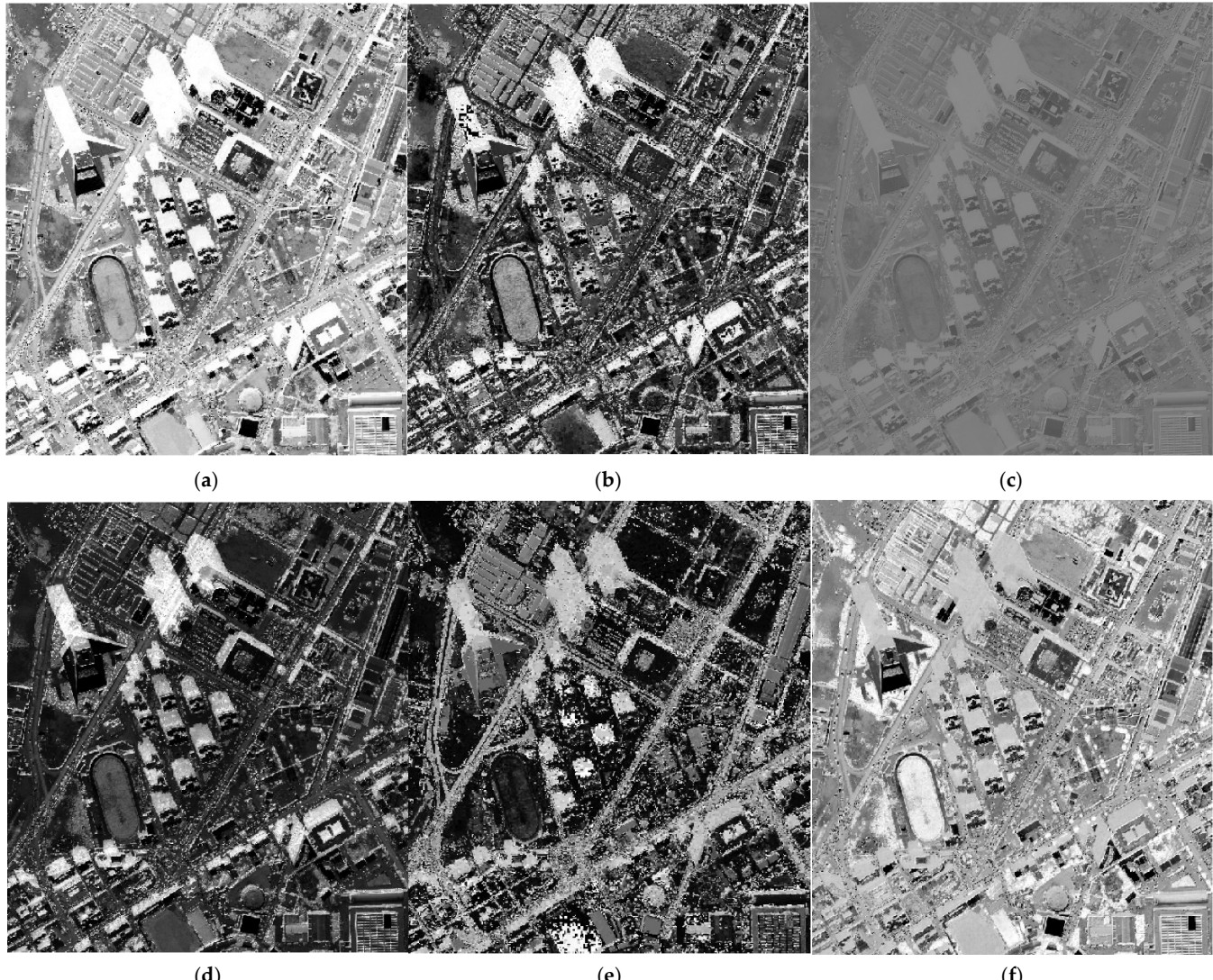

**Figure 11.** Comparisons of the indexes in Site 1: (**a**) ISI; (**b**) NSVDI; (**c**) CSI; (**d**) $T_{YCbCr}$; (**e**) $T_{HSV}$; and (**f**) SDI.

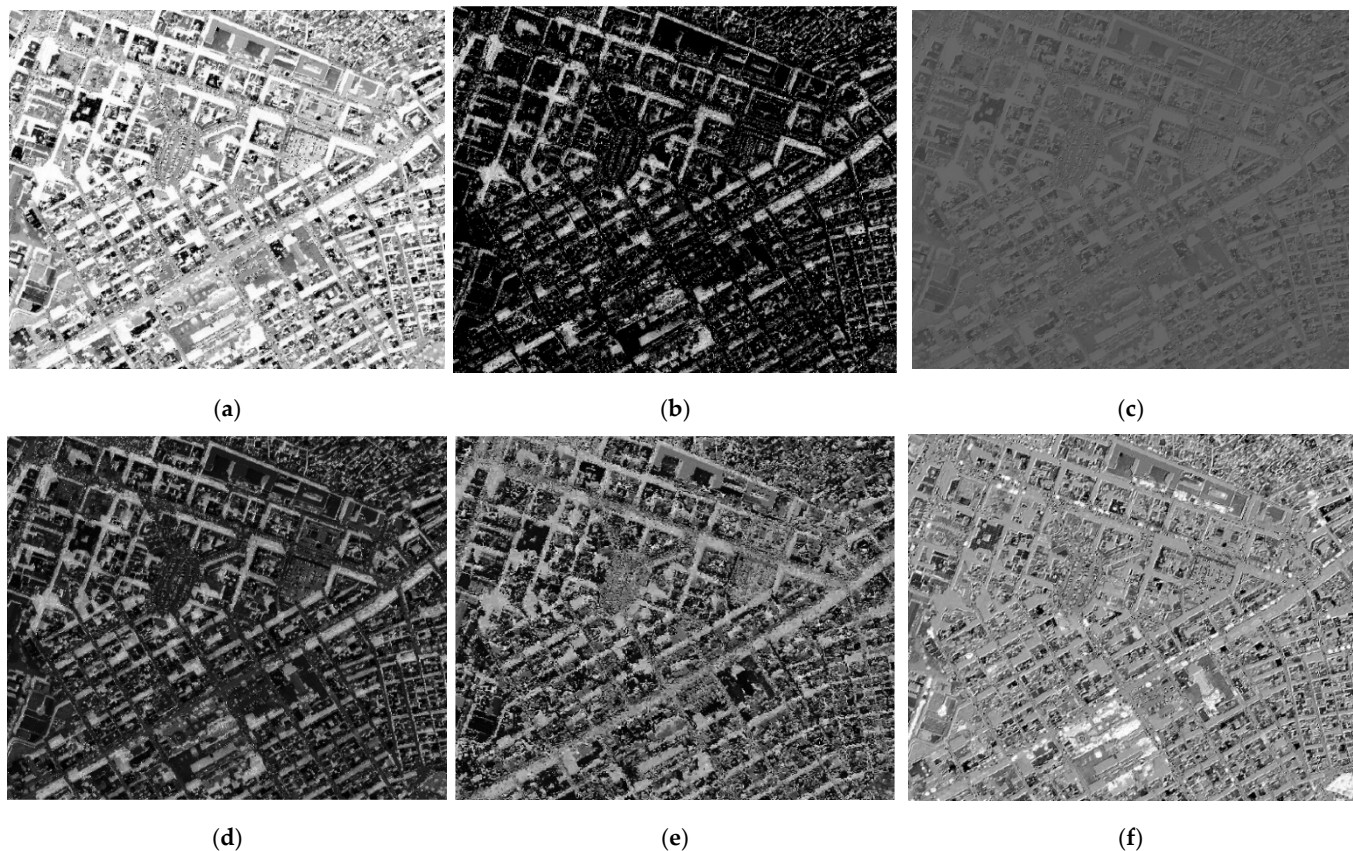

**Figure 12.** Comparisons of the indexes in Site 2. Comparisons of the indexes in Site 2: (**a**) ISI; (**b**) NSVDI; (**c**) CSI; (**d**) $T_{YCbCr}$; (**e**) $T_{HSV}$; and (**f**) SDI.

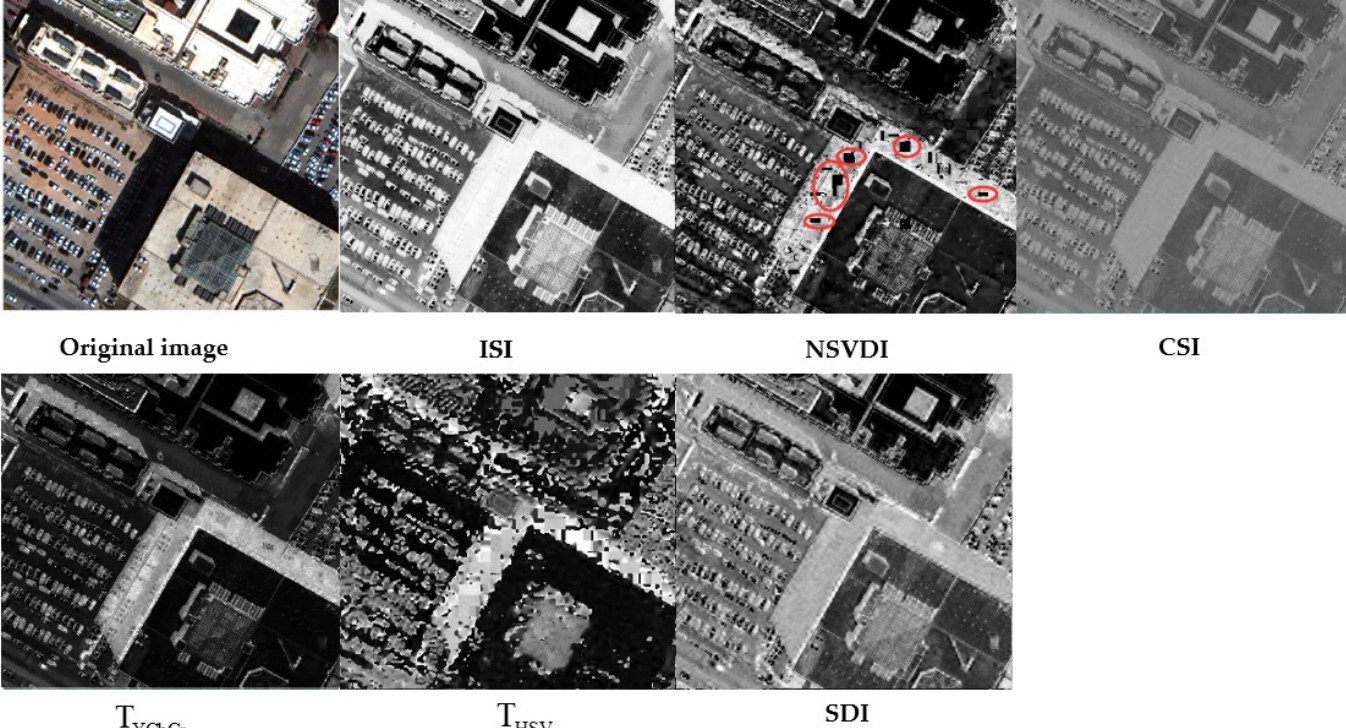

**Figure 13.** Enlarged details of each index result in Site 2: the area marked by the red ellipses in NSVDI are invalid values.

For the quantitative analysis of each index, we performed threshold segmentation on the calculation results of each index of Site 1 and Site 2, and evaluated the accuracy of the segmentation results. Among them, for the results of ISI, we first reconstructed it with Equation (4), and performed threshold segmentation on the reconstruction results. For the other five indexes, the thresholds were adjusted until the optimal shadow extraction results were obtained.

For the shadow detection results, the producer's accuracy (PA), user's accuracy (UA), overall accuracy (OA), and kappa coefficient (Kappa) from the confusion matrix [45] were employed to evaluate the accuracy of the shadow extraction, where OA and PA were used to evaluate the accuracy of the shadow detection, and the OA and the Kappa coefficient to assess the accuracy of discrimination between the shadow and the background:

$$
\begin{aligned}
PA &= TP/(TP+FN) \\
UA &= TP/(TP+FP) \\
OA &= (TP+TN)/N
\end{aligned}
\tag{16}
$$

where $TP$ (true positive) is the number of shadow pixels identified as shadow; $FN$ (false negative) is the number of shadow pixels but misjudged as non-shadow; $FP$ (false positive) is the number of non-shadow pixels but misjudged as shadow; $TN$ (true negative) is the number of non-shadow pixels identified as non-shadow; and $N = TP + TN + FN + FP$.

The shadow extraction accuracy of each index in Site 1 and Site 2 is listed in Tables 3 and 4. For Site 1, we know that the PAs of ISI, CSI and $T_{YCbCr}$ are all high, but CSI has a low UA (71%). This can be explained by the difficulty for CSI to distinguish between shadow and low reflectance areas, e.g., CSI mistakes asphalt roads as shadow. However, the accuracies of CSI at Site 2 are contrary to those of Site 1 (UA 95% and PA 86%), which is attributed to the fact that Site 2 is less affected by the low reflectance asphalt roads, but the landcover types are complex, resulting in many shadow areas that are not recognized. The accuracies of the shadow detection results of the $T_{YCbCr}$ both in Site 1 and Site 2 are considerably high, but still lower than ISI. The reason is that, although the $T_{YCbCr}$ can enhance shadow and distinguish it from the background, the enlarged figure in Figure 13 reveals that $T_{YCbCr}$ remains affected by the high reflectance landcovers in the shadow region. Both NSVDI and $T_{HSV}$ are affected by invalid values, so their accuracies are not desirable. The performance of SDI is the worst in both Site 1 and Site 2, and the results in Figures 11–13 show that SDI can neither distinguish low reflectance landcovers from shadow nor distinguish shadow from vegetation. In general, ISI provides the best performance in both experimental sites.

**Table 3.** Shadow extraction accuracy of the indexes in Site 1.

| | Accuracies | | | |
|---|---|---|---|---|
| Index | PA (%) | UA (%) | OA (%) | Kappa |
| ISI | 99 | 97 | 99 | 0.97 |
| NSVDI | 83 | 77 | 87 | 0.70 |
| CSI | 99 | 71 | 88 | 0.74 |
| $T_{YCbCr}$ | 96 | 90 | 95 | 0.89 |
| $T_{HSV}$ | 89 | 66 | 83 | 0.63 |
| SDI | 83 | 47 | 67 | 0.35 |

**Table 4.** Shadow extraction accuracy of the indexes in Site 2.

| Index | Accuracies | | | |
|---|---|---|---|---|
| | PA (%) | UA (%) | OA (%) | Kappa |
| ISI | 96 | 97 | 98 | 0.95 |
| NSVDI | 87 | 95 | 91 | 0.82 |
| SCSI | 86 | 95 | 90 | 0.81 |
| $T_{YCbCr}$ | 90 | 97 | 93 | 0.87 |
| $T_{HSV}$ | 75 | 89 | 81 | 0.63 |
| SDI | 69 | 80 | 74 | 0.47 |

*4.2. Assessment Shadow Compensation and Penumbra Removal*

4.2.1. Preference of Shadow Compensation and Penumbra Removal

In this section, we verified the effectiveness and feasibility of the proposed shadow compensation method and DPCM under complex urban conditions. Two experimental data from Site 1 and Site 2 were selected. Experimental data 1 mainly include the huge shadows, and the landcovers in the shadow area are complex with serious information loss, which impedes recovering shadow information. Experimental data 2 have dense buildings and are seriously affected by small noises, which is difficult for shadow extraction and beneficial to comprehensively observe shadow extraction accuracy and the shadow information compensation effect. After extracting the shadow using the shadow extraction method proposed in Section 2, we performed the shadow compensation and penumbra removal on the shadow area, and the results are shown in Figures 14 and 15. We can see that the shadow information of the two experimental data has been effectively restored, but in Figure 14b, it is obvious that the shadow edge is "oversaturated". As shown of the red rectangle in Figure 14b,c, after processing with DPCM, the penumbra has been accurately defined, and the information of the penumbra has been precisely restored. Figures 14c and 15c show that the shadows of large areas of complex backgrounds have been well removed. Compared with the original image, the features of various landcovers (e.g., vehicles, roads, vegetation) in the shadow areas are clearly recovered.

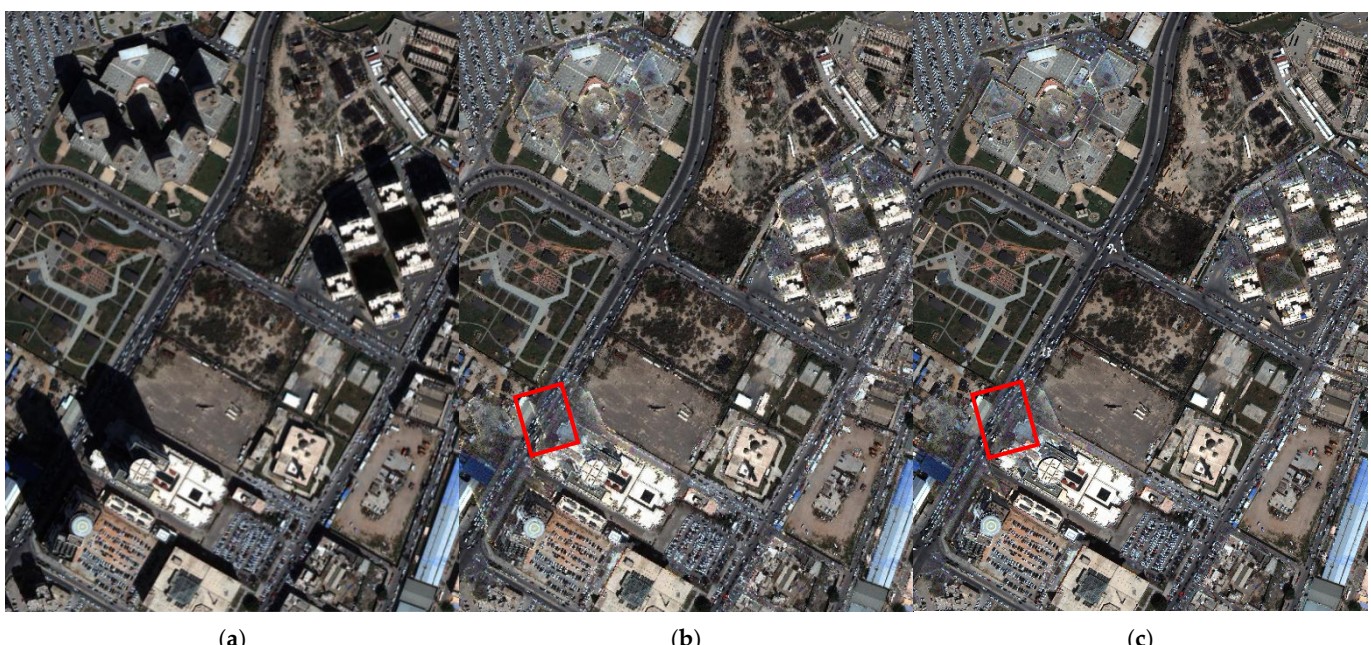

(**a**)　　　　　　　　　　(**b**)　　　　　　　　　　(**c**)

**Figure 14.** Results of experimental data 1: (**a**) original image; (**b**) result of shadow compensation; and (**c**) result of penumbra removal.

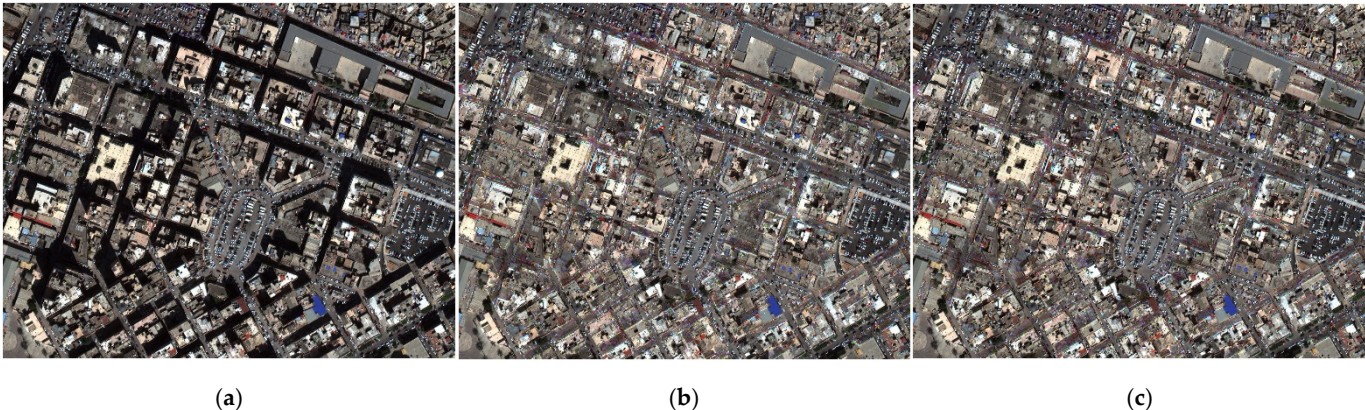

|  (**a**)  |  (**b**)  |  (**c**)  |

**Figure 15.** Results of experimental data 2: (**a**) original image; (**b**) result of shadow compensation; and (**c**) result of penumbra removal.

### 4.2.2. Comparative Analysis of Shadow Compensation

In order to verify the superiority of the shadow compensation algorithm proposed in this paper, the results of experimental data 1 and data 2 were compared with the most commonly used shadow removal algorithms. These methods are: the pyramid-based shadow removal method proposed by Shor et al. [46]; the paired regions method presented by Guo et al. [35]; the color-lines-related method by Yu et al. [47]; and the shadow terminator proposed by Bauer et al. [38]. The shadow removal results are shown in Figures 16–18. Moreover, in order to observe the details of shadow information recovery, we enlarged the results of experimental data 1 of each method, as exhibited in Figure 17.

Current shadow compensation algorithms are mainly aimed at a single simple image or a single shadow. The selected comparison methods have achieved good results in their own experimental data, but they are primarily applicable to restoring simple shadow information. As can be seen from the results in Figures 16–18, these methods generally provide poor effects in the shadow removal of remote sensing images with multiple shadows. For Shor's method, although shadows are detected correctly, only part of shadow information is effectively recovered (see Figure 17c), and many small noises still exist in these compensation results. As a computationally intensive process, Guo's method is not applicable for shadow compensation with multiple shadows, which may increase computational complexity and impede similar objects matching. As shown in Figure 16e to Figure 18e, the majority of the shadow objects are compensated with the unmatched non-shadow information. Yu's preference is second only to the proposed method, but it requires manual interaction with the input of shadow samples, non-shadow, and penumbra, which complicates not only the shadow process, but also makes it difficult to represent the whole situation through local samples for multiple shadows. Figures 16f and 18f illustrate that both shadow extraction accuracy and the preference of shadow compensation of Bauer's method are the worst among the five methods. Most of the low reflectance landcovers and vegetation are misjudged as shadows, which results in shadow compensation processed in these non-shadow regions, largely deteriorating the overall shadow compensation effect.

From the detailed diagram of the proposed method in Figure 18b, the shadow compensation effect of the proposed method is the optimal among the five methods. Roads, vehicles, buildings and bare soil which were originally blocked by shadows can be recognized by the naked eye after shadow compensation. However, due to the distance from the satellite sensor to the ground and the influence of aerosols, as well as the wide range of light intensity variation, the shadow removal results of the method presented are still with small noises, and information loss remains in the shadow compensation results, compared with the surrounding non-shadow regions.

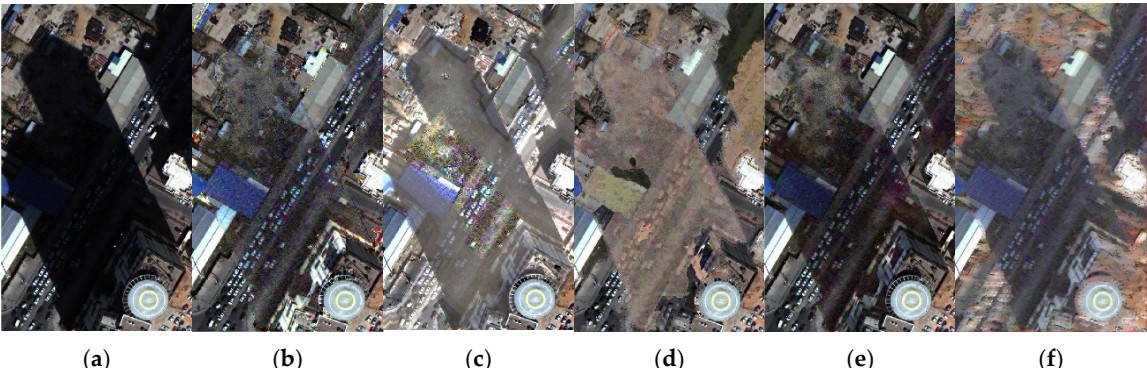

**Figure 16.** Shadow removal comparison of experimental data 1: (**a**) original image; (**b**) proposed; (**c**) Shor's; (**d**) Guo's; (**e**) Yu's; and (**f**) Bauer's.

**Figure 17.** Shadow removal details of the methods in experimental data 1: (**a**) original image; (**b**) proposed; (**c**) Shor's; (**d**) Guo's; (**e**) Yu's; and (**f**) Bauer's.

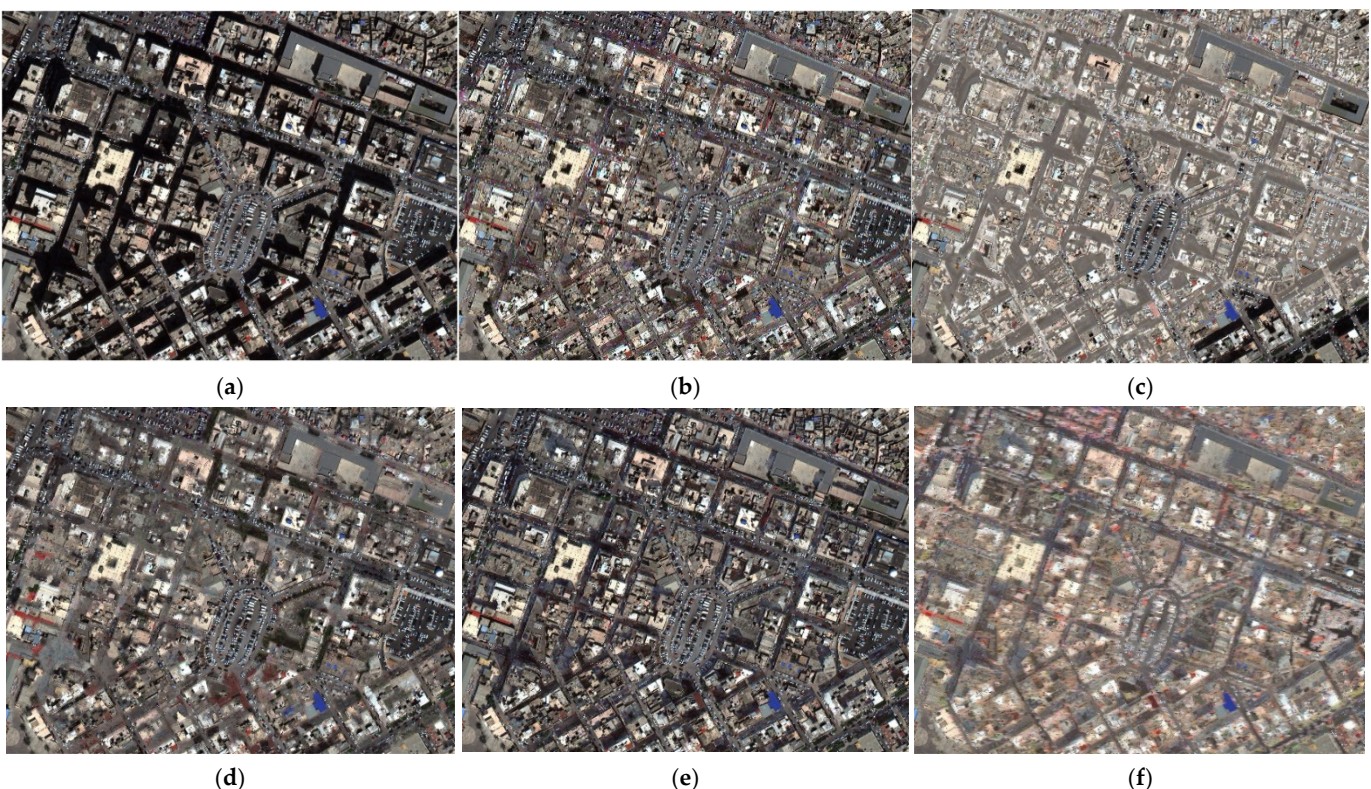

**Figure 18.** Shadow removal comparison of experimental data 2: (**a**) original image; (**b**) proposed; (**c**) Shor's; (**d**) Guo's; (**e**) Yu's; and (**f**) Bauer's.

#### 4.2.3. Comparative Analysis of Penumbra Removal

Many shadow removal methods only involve edge smoothing/matting, without defining and processing the penumbra separately, but this edge smoothing/matting is essentially penumbra removal. In consideration of the difficulty in obtaining the ground truth image of the remote sensing image, this paper selected the experimental images from the UIUC [35] dataset and SRD [23] dataset for testing, and compared the results of DCPM with Finlayson's [37], Bauer's [38] and Guo's [35] methods. Practically, inaccurate shadow detection would lead to incorrect penumbra information and wrong penumbra recovery, and comparison among the penumbra removal methods would be meaningless. Therefore, we adopted the same method for shadow extraction and compensation, and only make a comparative analysis for penumbra removal results. Considering no NIR band in the image from the datasets, SI was used to enhance the information of the shadow area, and then the proposed shadow detection and compensation methods are applied to obtain the shadow compensation results. We selected five images for testing, and the experimental results are shown in Figure 19. To better observe the details of the results of each method, we partially enlarged Figure 19, as shown in Figure 20.

It can be seen from the result graph in Figure 19, compared with the results of shadow compensation (without penumbra removal), the four methods can recover the penumbra information to varying degrees. However, the penumbra is a specific region where the light intensity changes dynamically. Bauer's method compensated all the penumbra region with the same strategy. Although the penumbra was removed, the information in the penumbra was still seriously lost. Finlayson's method used a gradient change method to optimize the edges, but since the penumbra was not specifically defined, the effect of penumbra restoration varied sharply on different data. From Figure 19, Guo's method appears to have achieved desirable results, but from the enlarged view of Figure 20, it also misses information in the penumbra region. The failure of these methods to effectively remove penumbra can be mainly ascribed to that they classify penumbra recovery as matting/edge

smoothing without an accurate definition of penumbra. For the proposed DPCM, good results have been achieved for all five images. From the detailed view in Figure 20, DPCM not only defines the penumbra range but also restores the information in penumbra in an accurate manner.

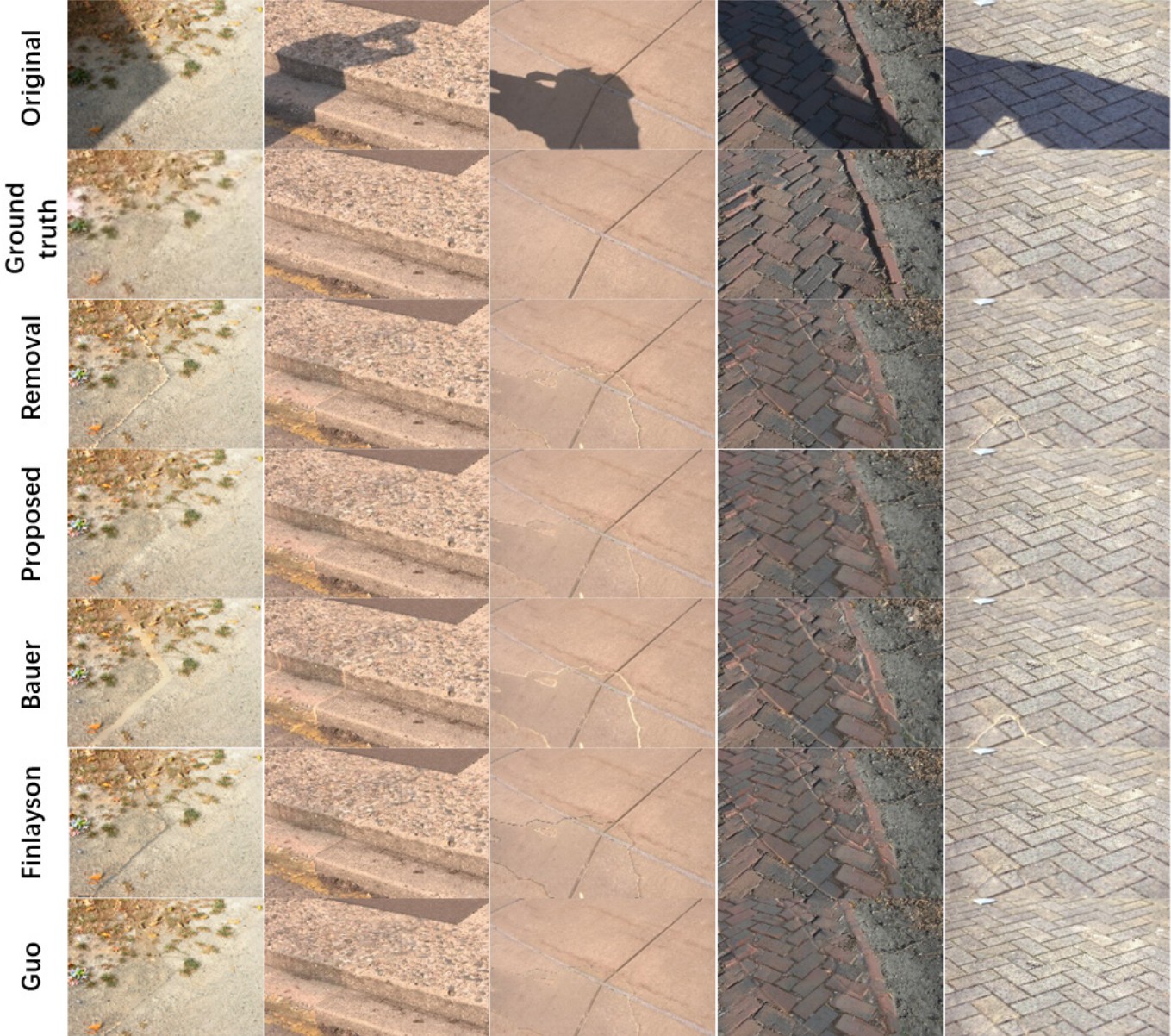

**Figure 19.** Comparison among the different penumbra removal methods. The first row is the original images; the second row is the ground truth; the third row is the results without penumbra removal; the fourth row is the results of the proposed method; the fifth row is the results of Bauer's method; the sixth row is the results of Finlayson's method; and the seventh row is the results of Guo's method.

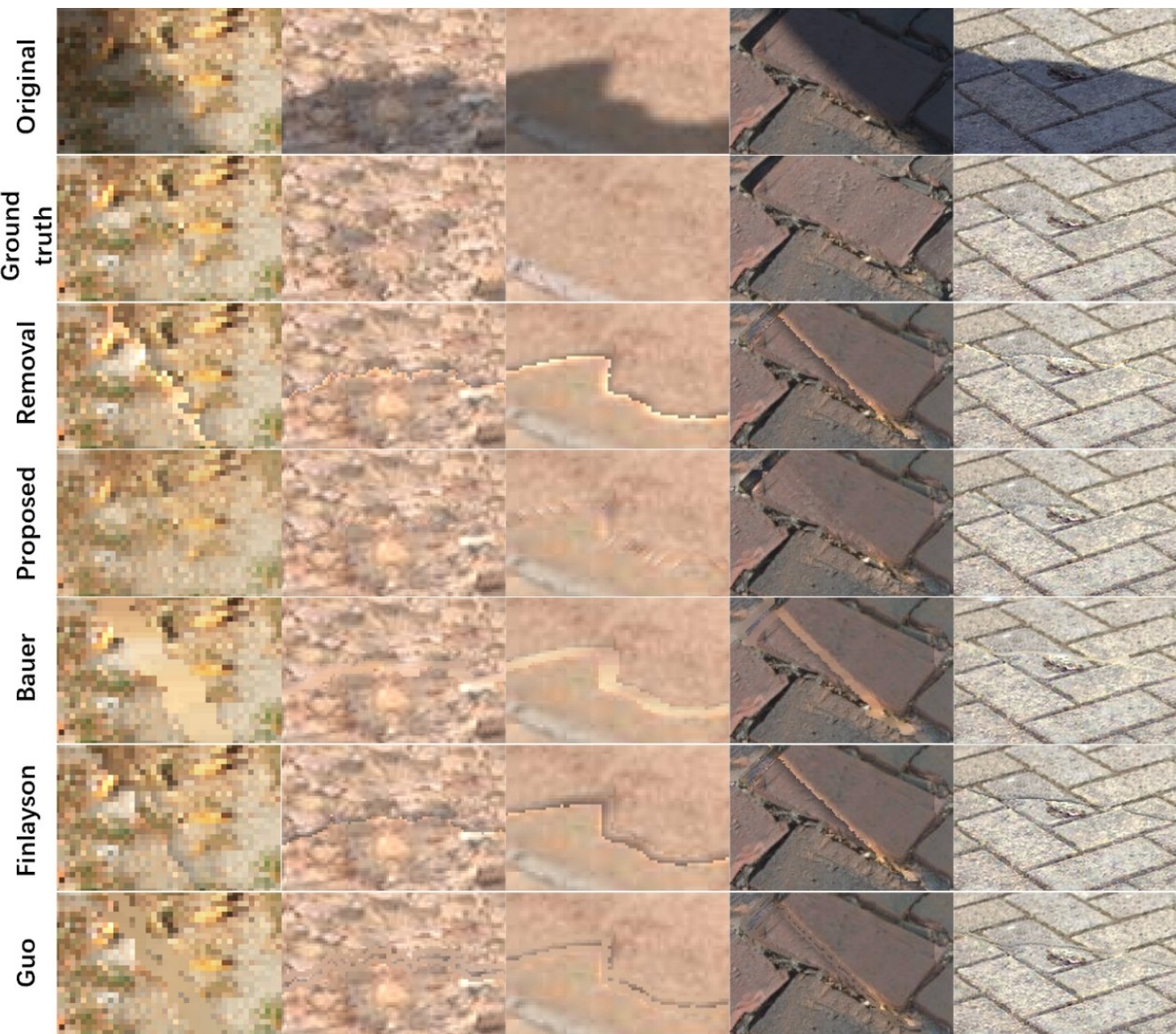

**Figure 20.** Details of the penumbra removal methods. The first row is the original images; the second row is the ground truth; the third row is the results without penumbra removal; the fourth row is the results of the proposed method; the fifth row is the results of Bauer's method; the sixth row is the results of Finlayson's method; and the seventh row is the results of Guo's method.

## 5. Discussion

Current shadow removal algorithms, especially the developing deep-learning models of recent years, provide new ideas for shadow removal. The application of deep learning for shadow removal has made accomplishments, but the current studies are mainly concerned with the removal of ordinary images or single shadow. The studies of shadow on remote sensing images primarily focus on clouds and cloud shadow removal, while the development of shadows formed by ground objects blocking remains immature. At present, the related applications of urban remote sensing are extremely common, but the information loss is serious in remote sensing images in urban areas due to the light blocked by tall buildings and viaducts. The accurate removal of these shadows is an important prerequisite for urban remote sensing-related research such as the three-dimensional reconstruction of buildings as well as the classification of urban features. Despite the satisfactory results in ordinary images achieved by the deep learning of shadow removal,

for remote sensing images, the influence of light changes and revisits cycles of satellite hinder obtaining training dataset in the same site.

The purpose of this paper is to design a method for shadow extraction and information restoration suitable for remote sensing image with multiple shadows and complex surface in urban areas. For shadow detection, we improved the existing shadow index, and used image reconstruction to enhance the completeness and accuracy of shadow detection. For shadow compensation, we designed a new scheme combining the image segmentation objects with the information of non-shadow objects adjacent to the shadow, so as to compensate the shadow information. We also proposed DCPM to accurately remove the penumbra. The process and analysis of this work are specified as follows:

After analyzing the existing color space, learning that YCbCr space was more sensitive to shadows, we proposed SI to enhance shadows, and when combined with the NIR band, weakened the influence of high reflectance landcovers in the shadow area. During shadow extraction, we reconstructed the ISI image, and performed threshold segmentation on the reconstructed image to obtain the shadow mask. In this way, we can reduce the influence of the small shadows formed by vehicles and other small landcovers as well as noises, and optimize the integrity of the shadow edge. Aiming at verifying the effectiveness and accuracy of the proposed shadow detection method, we selected two sites to test ISI, and made a qualitative analysis of ISI and state-of-art indexes. In addition, we conducted threshold segmentation for reconstructed ISI and other indexes, and performed a quantitative analysis of the shadow extraction accuracy using UA, PA, OA and Kappa. The analysis results demonstrate that the anti-noise ability of ISI and the integrity and accuracy of shadow extraction are higher than those of the other four indexes. The performance ranking of the five indexes is: $ISI > T_{YCbCr} > CSI > NSVDI > ISI > T_{HSV} > SDI$.

For the shadow compensation part, this paper compensated the shadow information of RGB bands. We took the objects from the mean-shift segmentation as the smallest processing unit and used the object-based method to compensate the shadow objects. Knowing that each object has the most similar illumination conditions to its neighbors, we adopted the information from non-shadow objects adjacent to the shadow objects to restore the information of the shadow objects. For the shadow compensation result, we also selected two experimental data for testing, and compared them with the existing shadow removal methods. The results show that the performance of the proposed method ranks the best among the five methods. The worse effects of the other four may be attributed to the inaccurate shadow detection results or the drawbacks of the shadow compensation strategy. Moreover, Guo's method and Yu's method have achieved satisfactory results in removing simple shadows in their own dataset, but they are not suitable for the shadow removal of remote sensing images with multiple shadows and complex landcovers.

From the transition area between non-shadow and shadow, the light intensity consists of ambient light and part of the direct light and is accompanied by the gradual attenuation of the direct light intensity, which is called penumbra. After statistical analysis, we defined the penumbra range and proposed the DPCM to removal penumbra dynamically. To observe the effect of penumbra removal more intuitively, we used a dataset containing both shadows and non-shadows for testing and compared them with other three edge smoothing/matting methods. The comparison results manifest that DPCM has the best effect of the four methods.

## 6. Conclusions

To solve the problem that remote sensing images in urban areas are susceptible to shadow occlusion, this paper proposes an effective method to remove shadows from remote sensing images with multiple shadows and complex backgrounds. For shadow detection, we proposed ISI to enhance the shadow and reconstructed ISI to weaken the interference of noise and improve the integrity. Since adjacent objects are likely to have the same ambient light intensity, a shadow compensate technique was implemented to recover the shaded objects with the unshaded objects adjacent to them. For the problem of "oversaturation" on

the shadow compensation edge, a dynamic penumbra compensation method (DPCM) was presented to define the penumbra region and dynamically remove the penumbra. For the purpose of verifying the robustness and effectiveness of the proposed method, the results of shadow extraction, shadow compensation and penumbra removal were analyzed and compared with those of the state-of-art methods. The experimental results show that the proposed method was applicable for the shadow removal of remote sensing images and beneficial for the development and application of urban remote sensing.

**Author Contributions:** Conceptualization of the experiments: T.Z., H.F. and C.S.; data curation: T.Z.; methodology: T.Z., and H.F.; funding acquisition: C.S. and S.W.; data analysis: T.Z. and H.F.; writing—original draft preparation: T.Z. and H.F.; writing—review and editing: H.F. and C.S.; supervision: H.F., C.S. and S.W. All authors have read and agreed to the published version of the manuscript.

**Funding:** This work is supported by National Natural Science Foundation of China (No. 11104106, 11574113, 11374123), Science and Technology Planning Project of Jilin Province (No. 20170204076GX, 20180101006JC, 20180101238JC, 20190103041JH, 20190201260JC, 20190201306JC, 20200201179JC, 2019C0355-5, JJKH20200935KJ), as well as supported by Graduate Innovation Fund of Jilin University (101832020CX078).

**Institutional Review Board Statement:** Not applicable.

**Informed Consent Statement:** Informed consent was obtained from all subjects involved in the study.

**Data Availability Statement:** Data available in a publicly accessible repository. The data presented in this study are openly available on http://www.digitalglobe.com/samples (accessed on 9 December 2020), reference number [40].

**Acknowledgments:** The authors would like to thank the editors end reviewers for their suggestions and revisions. For the data used in this research, the WorldView-3 image was provided thanks to the courtesy of the DigitalGlobe. The authors are also highly grateful for the help from the excellent translator (Yun Nie).

**Conflicts of Interest:** The authors declare no conflict of interest.

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
