# Peer review of "Shadow Detection and Compensation from Remote Sensing Images under Complex Urban Conditions"

_remotesensing, doi:10.3390/rs13040699_

Round 1

Reviewer 1 Report

The authors present a new shadow detection and mitigation approach for high resolution RGB imagery. The paper presents some nice examples of the approach but is difficult at times to keep all of the different approaches clearly organized.  The authors have improved SI by adding in an NIR band to form the ISI algorithm. Some of the images show the SI approach and others show the ISI approach or both approaches. This is a little confusing for the reader and may have led to a mislabeling of the second panel of Figure 13. Always showing the SI and the ISI images may be the best solution.

The first row of Figure 3 is missing, so this whole section was not very clear.

Finally, many of the spatial parameters are described in terms of pixels, but shouldn't the spatial resolution of the imager play a role in setting the parameters? This issue is particularly important in the mean-shift segmentation discussion.

From a technical perspective I wonder how the approach would work if a pool of water was contained or partly contained in a deep shadow.  Water is very dark and many algorithms incorrectly identify it as a shadow.

There are also occasional sentences which are a bit awkward and tend to confuse the discussion.

Author Response

Dear reviewer:

Firstly, thank you very much for your comments concerning our manuscript. Those comments are all valuable and very helpful for revising and improving our paper, as well as the important guiding significance to our researches. We have modified each suggestion accordingly, and I believe the article has been comprehensively improved. We very much hope that our revision could satisfy you. If there are any other mistakes or improper writing, please inform us and we will modify carefully. Finally, look forward to hearing from you soon.

Point 1: The authors present a new shadow detection and mitigation approach for high resolution RGB imagery. The paper presents some nice examples of the approach but is difficult at times to keep all of the different approaches clearly organized. The authors have improved SI by adding in an NIR band to form the ISI algorithm. Some of the images show the SI approach and others show the ISI approach or both approaches. This is a little confusing for the reader and may have led to a mislabeling of the second panel of Figure 13. Always showing the SI and the ISI images may be the best solution.

Response 1: Thank you very much for your reminding, in Section 4.1.2, we made a comparative analysis for the indexes. We post the result of SI in Figure 13 but not in Figure 11 and Figure 12, which really makes the reader difficult to understand. Therefore, we deleted the results of SI in Figure 13.

In addition, we are so sorry that we mistakenly write ISI as NSI in Table 4 and Table 5 and other places, so that readers cannot find the shadow extraction accuracy of ISI in Table 4 and Table 5. Besides, SI is our intermediate process of designing ISI. The results of SI are shown in Figure 9 and Figure 10 to illustrate that ISI can better reduce the interference of high-reflectance objects in the shadow area compared with SI, which can be better observed in Figures 3f and g

Point 2: The first row of Figure 3 is missing, so this whole section was not very clear.

Response 2: We are very sorry for the mistake, we forgot to paste Figure 3 (a-e), and we have revised it.

Point 3: Finally, many of the spatial parameters are described in terms of pixels, but shouldn't the spatial resolution of the imager play a role in setting the parameters? This issue is particularly important in the mean-shift segmentation discussion.

Response 3: Thank you for your suggestion, and we have also added a supplementary explanation in the corresponding position of the manuscript. When using the Mean-shift algorithm, the spatial resolution of image is proportional to the parameter size setting of Mean-shift. Highly reflectance landcovers such as white vehicles, etc., interfere with shadow extraction. If these high reflectance landcovers can be averaged with other surrounding shadow pixels, the impact of high reflectance landcovers on shadow extraction will be greatly reduced. In the remote sensing image of urban areas, high reflectance vehicles and small noises are the main cause of obvious interference to shadow detection. Taking WorldView-3 data as an example, the size of the vehicle is about 4.5m*1.5m (the spatial resolution of the image is 0.31m, one vehicle occupies about 75 pixels), when we set the minimum segment area above 150, the high reflectance vehicle will be merged with the surrounding shadow pixels into one object, and the impact of high reflectance vehicle will be reduced. After comparative analysis, we set segmentation spatial radius as 9, segmentation feature space radius as 15 and minimum segment area as 200 in the mean-shift.

Point 4: From a technical perspective I wonder how the approach would work if a pool of water was contained or partly contained in a deep shadow. Water is very dark and many algorithms incorrectly identify it as a shadow.

Response 4: Your question is very meaningful, and we have considered it in our previous work. If the water body is outside the shadow, the NDWI of water body will be greater than that of shadow in most cases. We found that when the light intensity is strong or the image has abundant bands, NDWI can distinguish between shadows and water bodies. However, if the light intensity is very weak and the image has only four bands of R, G, B and NIR or less, then the water body and the shadow cannot be distinguished.

In practical applications, if the water body is outside the shadow, the water body can be masked by using NDWI. If the water is in the shadow, then this part of the area will still be judged as shadow, and the proposed compensation method will be used in this region.  However, we think it is very difficult to extract the water body covered by shadows. In this paper, we did not consider the impact of water bodies, but our research on this issue is still going on.

Point 5: There are also occasional sentences which are a bit awkward and tend to confuse the discussion.

Response 5: We are so sorry for our unprofessional English writing! After carefully check, we have checked it carefully and modified the manuscript accordingly. If there are still any problems, please tell us and we will modify timely.

Reviewer 2 Report

The research has been well-planned and executed. Detailed information on the methods employed has been given. Moreover, the results and discussion section ae appropriate.

There are two minors’ issues that need to be taken care of:

  • Between lines 127-129; the authors claim that current shadow detection methods are not suitable for remote sensing image in large scale. However, they do not define what large scale represent, nor address the issue in the research. They should either show how their research is applicable to large area; or remove the limitation they highlighted.
  • The second contribution of the paper is “the restauration of shadow objects information through nearest non-shadow objects” (L139-140). There is an assumption here that adjacent objects are of similar nature. This assumption needs a good justification.

Other notes:

L72: To strengthen

L78: it is difficult to it is difficult: Awkward sentence 

L150-152; L270: we put forward a shadow compensation method which always compensate shadow objects with their adjacent non-shadow objects since the adjacent objects are most likely to have the same material and ambient light intensity. This assumption is not warranted all the time. Please justify it.

L161-162: “face of objects when it not directly illuminated by sunlight”: Awkward sentence; reformulate.

Author Response

Dear reviewer:

Firstly, thank you very much for your comments concerning our manuscript. Those comments are all valuable and very helpful for revising and improving our paper, as well as the important guiding significance to our researches. We have modified each suggestion accordingly, and I believe the article has been comprehensively improved. We very much hope that our revision could satisfy you. If there are any other mistakes or improper writing, please inform us and we will modify carefully. Finally, look forward to hearing from you soon.

The responses are as follows:

Point 1: Between lines 127-129; the authors claim that current shadow detection methods are not suitable for remote sensing image in large scale. However, they do not define what large scale represent, nor address the issue in the research. They should either show how their research is applicable to large area; or remove the limitation they highlighted.

Response 1: Thank you very much for your suggestion. At present, most of the methods of shadow information recovery are designed for single shadows, such as the shadow cast by a building. The method proposed in this paper is suitable for multiple shadows to perform information compensation at the same time. We are sorry that we did not have an accurate definition of large scale. After consideration, we have changed the large scale to multiply shadows in the corresponding position of the manuscript.

Point 2: The second contribution of the paper is “the restauration of shadow objects information through nearest non-shadow objects” (L139-140). There is an assumption here that adjacent objects are of similar nature. This assumption needs a good justification.

Response 2: Thank you for your correction. It is not rigorous to describe that adjacent objects have similar materials. We calculated the similarity of ambient light and direct light during the shadow compensation without considering the similar of material. Therefore, we have deleted all the descriptions in the manuscript that adjacent objects may have the same material composition.

Point 3:

L72: To strengthen

L78: it is difficult to it is difficult: Awkward sentence

Response 3: Thanks for pointing out the mistakes. We have modified the in the corresponding position of the manuscript

Point 4: L150-152; L270: we put forward a shadow compensation method which always compensate shadow objects with their adjacent non-shadow objects since the adjacent objects are most likely to have the same material and ambient light intensity. This assumption is not warranted all the time. Please justify it.

Response 4: Thank you for reminding, our description of adjacent objects having the same material is not accurate. We have modified them in the corresponding positions of the manuscript.

Point 5: L161-162: “face of objects when it not directly illuminated by sunlight”: Awkward sentence; reformulate.

Response 5: We have modified it in the corresponding positions of the manuscript.
